# Convergence of an actor-critic gradient flow for entropy regularised MDPs in general spaces

**Denis Zorba, David Šiška & Lukasz Szpruch**
School of Mathematics
University of Edinburgh
Edinburgh, United Kingdom
{e.zorba, D.Siska, L.Szpruch}@ed.ac.uk

## Abstract

We prove the stability and global convergence of a coupled actor-critic gradient flow for infinite-horizon and entropy-regularised Markov decision processes (MDPs) in continuous state and action space with linear function approximation under Q-function realisability. We consider a version of the actor critic gradient flow where the critic is updated using temporal difference (TD) learning while the policy is updated using a policy mirror descent method on a separate timescale. For general action spaces, the relative entropy regularizer is unbounded and thus it is not clear a priori that the actor-critc flow does not suffer from finite-time blow-up. Therefore we first demonstrate stability which in turn enables us obtain a convergence rate of the actor critic flow to the optimal regularised value function. The arguments presented show that timescale separation is crucial for stability and convergence in this setting.

## 1 Introduction

In reinforcement learning (RL) an agent aims to learn an optimal policy that maximizes the expected cumulative reward through repeated interactions with its environment. Such methods typically involve two key components: policy evaluation and policy improvement. During policy evaluation, the advantage function corresponding to a policy, or its function approximation, is updated using state, action and reward data generated under this policy. Policy improvement then uses this approximate advantage function to update the policy, most commonly through some policy gradient method. Algorithms that explicitly combine these two components are known as actor-critic (AC) methods Konda & Tsitsiklis (1999), where the actor corresponds to policy improvement and the critic to policy evaluation.

There are many policy gradient methods to choose from. In the last decade trust region policy optimization (TRPO) methods Schulman et al. (2015) and methods inspired by these like PPO Schulman et al. (2017) have become increasingly well-established due to their impressive empirical performance. Largely, this is because they alleviate the difficulty in choosing appropriate step sizes for the policy gradient updates: for vanilla policy gradient even a small change in the parameter may result in large change in the policy, leading to instability, but TRPO prevents this by explicitly ensuring the KL divergence between successive updates is smaller than some tolerance. Mirror descent replaces the TRPO's hard constraint with a penalty leading to a first order method which is also ameanable to analysis. Indeed, at least for direct parametrization, it is known to converge with sub-linear and even linear rate for entropy regularised problems (depending on exact assumptions) Ju & Lan (2024); Lan (2023); Kerimkulov et al. (2025a).

Due to the favourable analytical properties of mirror descent, in this paper we consider a version of the actor critic gradient flow where the policy is updated using a policy mirror descent method while the critic is updated using temporal difference (TD) on a separate timescale.

Entropy-regularised MDPs are widely used in practice since the entropic regularizer leads to a number of desirable properties: it has a natural interpretation as something that drives exploration, it

ensures that there is a unique optimal policy and it can accelerate convergence of policy gradient methods Mei et al. (2020). However, analysing the stability and convergence of actor-critic methods in this entropy-regularised setting with general state and action spaces remains highly non-trivial due to lack of a priori bounds on the value functions.

To address the actor critic methods for entropy regularised MDPs in general action spaces, a careful treatment of tools from two timescale analysis, convex analysis over both Euclidean spaces and measure spaces must be deployed.

In this paper, we address precisely this challenge. We study the stability and convergence of a widely used actor-critic algorithm in which the critic is updated using Temporal Difference (TD) learning Sutton (1988), and the policy is updated through Policy Mirror Descent Ju & Lan (2024). Our analysis employs a two-timescale update scheme Borkar & Konda (1997), where both the actor and critic are updated at each iteration with the critic updated on a faster timescale.

## 1.1 RELATED WORKS

We focus on the subset of RL literature that address the convergence of coupled actor-critic algorithms. In the unregularised setting, actor-critic methods have been studied extensively. The first convergence results in the two-timescale regime established asymptotic convergence in the continuous-time limit of coupled updates (Borkar & Konda (1997); Konda & Tsitsiklis (1999)). Most modern research employs linear function approximation for the critic, where linear convergence rates have been obtained under various assumptions on the step-sizes of the actor and critic (Barakat et al. (2022); Zhang et al. (2020); Hong et al. (2023)).

Closely related to our work is Zhang et al. (2021), which considers the same two-timescale actor-critic scheme in the continuous-time limit for unregularised MDPs, with an overparameterized neural network used for the critic. However, convergence to the optimal policy was not established, and a restarting mechanism was required to ensure the stability of the dynamics.

In the entropy-regularised setting, Cayci et al. (2024a;b) address the convergence of a natural actor critic algorithm. However, the convergence and stability of these results rely on the finite cardinality of the action space in presence of entropy regularisation.

## 1.2 OUR CONTRIBUTION

Under linear $Q_\tau^\pi$-realisability assumption, we address the following question:

*"Is the actor-critic gradient flow for entropy-regularised MDPs in general action spaces stable and convergent, and if so, at what rate?"*

There are two main technical challenges one has to overcome when working with entropy-regularised MDPs in general action spaces.

- Even in mirror descent with exact advantage, the rate of convergence depends on a constant term $\int_S \mathrm{KL}(\pi^*(\cdot|s)|\pi_0(\cdot|s)d_\rho^{\pi^*}(ds)$. See Lan (2023); Kerimkulov et al. (2025a). In general action spaces, without entropy regularisation it is almost impossible to choose $\pi_0$ which would make this term finite, see Remark 2.1. Thus we need to include the regularisation in the analysis.

- Moreover, ensuring that the relative entropy does not blow up is difficult in general action spaces. In the finite action space setting, for any measure $\mu \in \mathcal{P}(A)$ such that $\mu(a_i) > 0$ and for all $s \in S$ it holds that $\mathrm{KL}(\pi(\cdot|s)|\mu) \leq \log|A|$. In general action spaces the KL divergence has no upper bound (can be $+\infty$) even if $\mu$ has full support. Under mild assumptions we show that the KL divergence does not blow up in finite time, see Corollary 5.1.

Our main contributions are as follows:

- We study a common variant of actor-critic where the critic is updated using temporal difference (TD) learning and the policy is updated using mirror descent. Similarly to Konda & Tsitsiklis (1999); Zhang et al. (2021), we analyse the coupled updates in the continuous-time limit, resulting in a dynamical system where the critic flow is captured by a *semi-*

gradient flow and the actor flow corresponds to an approximate Fisher–Rao gradient flow over the space of probability kernels.

- By combining convex analysis over the space of probability measures with classical Euclidean convex analysis, we develop a Lyapunov-based stability framework that captures the interplay between entropy regularisation and timescale separation, and establish stability of the resulting dynamics.

- We prove convergence of the actor-critic dynamics for entropy-regularised MDPs with infinite action spaces.

## 1.3 NOTATION

Let $(E, d)$ denote a Polish space (i.e., a complete separable metric space). We always equip a Polish space with its Borel sigma-field $\mathcal{B}(E)$. Denote by $B_b(E)$ the space of bounded measurable functions $f : E \to \mathbb{R}$ endowed with the supremum norm $|f|_{B_b(E)} = \sup_{x \in E} |f(x)|$. Denote by $\mathcal{M}(E)$ the Banach space of finite signed measures $\mu$ on $E$ endowed with the total variation norm $|\mu|_{\mathcal{M}(E)} = |\mu|(E)$, where $|\mu|$ is the total variation measure. Recall that if $\mu = f\, d\rho$, where $\rho \in \mathcal{M}_+(E)$ is a nonnegative measure and $f \in L^1(E, \rho)$, then $|\mu|_{\mathcal{M}(E)} = |f|_{L^1(E,\rho)}$. Denote by $\mathcal{P}(E) \subset \mathcal{M}(E)$ the set of probability measures on $E$. Let $\delta_x \in \mathcal{P}(E)$ denote the Dirac measure with mass at $x \in E$. Moreover, we denote the Euclidean norm on $\mathbb{R}^N$ by $|\cdot|$ with inner product $\langle \cdot, \cdot \rangle$. Given some $A, B \in \mathbb{R}^{N \times N}$, by $\lambda_{\min}(A)$ we denote the minimum eigenvalue of $A$ and write $A \succeq B$ if and only if $A - B$ is positive semidefinite. Finally, we denote by $|A|_{\mathrm{op}}$ the operator norm of $A$ induced by the Euclidean norm, $|A|_{\mathrm{op}} := \sup_{|x| \neq 0} \frac{|Ax|}{|x|}$.

## 1.4 ENTROPY REGULARISED MARKOV DECISION PROCESSES

Consider an infinite horizon Markov Decision Process $(S, A, P, c, \gamma)$, where the state space $S$ and action space $A$ are Polish, $P \in \mathcal{P}(S|S \times A)$ is the state transition probability kernel, $c$ is a bounded cost function and $\gamma \in (0, 1)$ is a discount factor. Let $\mu \in \mathcal{P}(A)$ denote a reference probability measure and $\tau > 0$ denote a regularisation parameter. Let $\rho \in \mathcal{P}(S)$ be an arbitrary initial state distribution. To ease notation, for each $\pi \in \mathcal{P}(A|S)$ we define the kernels $P_\pi(ds'|s) := \int_A P(ds'|s, a)\pi(da|s)$ and $P^\pi(ds', da'|s, a) := P(ds'|s, a)\pi(da'|s')$. Due to (Bertsekas & Shreve, 1996, Proposition 7.28) we can construct a probability measure $\mathbb{P}_\rho^\pi$, expectation $\mathbb{E}_\rho^\pi$ and stochastic processes $(s_n)_{n \in \mathbb{N}_0}$, $(a_n)_{n \in \mathbb{N}_0}$ with the conditional transition probabilities corresponding to those given by $P^\pi$ and $\pi$ respectively and with $s_0 \sim \rho$. Let $\mathbb{E}_s^\pi := \mathbb{E}_{\delta_s}^\pi$. For $\pi \in \mathcal{P}(A|S)$ define the regularised value function as

$$S \ni s \mapsto V_\tau^\pi(s) = \mathbb{E}_s^\pi \left[ \sum_{n=0}^\infty \gamma^n \Big( c(s_n, a_n) + \tau \, \mathrm{KL}(\pi(\cdot|s_n)|\mu) \Big) \right] \in \mathbb{R} \cup \{\infty\},$$

where $\mathrm{KL}(\pi(\cdot|s)|\mu)$ is the Kullback-Leibler (KL) divergence of $\pi(\cdot|s)$ with respect to $\mu$, define as $\mathrm{KL}(\pi(\cdot|s)|\mu) := \int_A \ln \frac{d\pi}{d\mu}(a|s)\pi(da|s)$ if $\pi(\cdot|s)$ is absolutely continuous with respect to $\mu$, and infinity otherwise.

For a given initial distribution $\rho \in \mathcal{P}(S)$, the optimal value function is defined as

$$V_\tau^*(\rho) = \min_{\pi \in \mathcal{P}(A|S)} V_\tau^\pi(\rho), \quad \text{with } V_\tau^\pi(\rho) := \int_S V_\tau^\pi(s)\rho(ds)$$

and we refer to $\pi^* \in \mathcal{P}(A|S)$ as the optimal policy if $V_\tau^*(\rho) = V_\tau^{\pi^*}(\rho)$. The Bellman Principle for entropy regularised MDPs, see Theorem A.1, suggests that without loss of generality, it is sufficient to consider policies from the class given by Definition 1.1 below.

**Definition 1.1** (Admissible Policies). *Let $\Pi_\mu$ denote the class of policies for which there exists $f \in B_b(S \times A)$ with*

$$\pi(da|s) = \frac{\exp(f(s, a))}{\int_A \exp(f(s, a))\mu(da)} \mu(da).$$

For each $\pi \in \Pi_\mu$ the value function $V_\tau^\pi$ is the unique bounded solution of the on-policy Bellman equation

$$V_\tau^\pi(s) = \int_A \left( Q_\tau^\pi(s, a) + \tau \ln \frac{d\pi}{d\mu}(a, s) \right) \pi(da|s), \tag{1}$$

see e.g. (Kerimkulov et al., 2025a, Lemma B.2).

For each $\pi \in \Pi_\mu$, we define the state-action value function $Q_\tau^\pi \in B_b(S \times A)$ by

$$Q_\tau^\pi(s, a) = c(s, a) + \gamma \int_S V_\tau^\pi(s') P(ds'|s, a). \tag{2}$$

We see that $Q_\tau^\pi : S \times A \to \mathbb{R}$ is a fixed point of $\mathrm{T}^\pi : B_b(S \times A) \to B_b(S \times A)$, defined as

$$\mathrm{T}^\pi f(s, a) = c(s, a) + \gamma \int_{S \times A} f(s', a') P^\pi(ds', da'|s, a) + \tau\gamma \int_S \mathrm{KL}(\pi(\cdot|s')|\mu) P(ds'|s, a). \tag{3}$$

As one can show this operator is a contraction, we see that $Q_\tau^\pi$ is in fact the unique fixed point.

## 2 MIRROR-DESCENT AND THE FISHER–RAO GRADIENT FLOW

Let the soft advantage function be defined as

$$A_\tau^\pi(s, a) := Q_\tau^\pi(s, a) + \tau \ln \frac{d\pi}{d\mu}(s, a) - V_\tau^\pi(s).$$

Then for some $\lambda > 0$ and $\pi_0 \in \Pi_\mu$, the Policy Mirror Descent update rule reads as

$$\pi^{n+1}(\cdot|s) = \arg\min_{m \in \mathcal{P}(A)} \left[ \int_A A_\tau^{\pi^n}(s, a)(m(da) - \pi^n(da|s)) + \frac{1}{\lambda} \mathrm{KL}(m|\pi^n(\cdot|s)). \right]$$

Due to (Dupuis & Ellis, 1997, Lemma 1.4.3) we know that this pointwise minimum is achieved by

$$\frac{d\pi^{n+1}}{d\pi^n}(a, s) = \frac{\exp\left(-\lambda A_\tau^{\pi^n}(s, a)\right)}{\int_A \exp\left(-\lambda A_\tau^{\pi^n}(s, a)\right) \pi^n(da|s)}. \tag{4}$$

From (1) we note that for any $\pi \in \mathcal{P}(A|S)$, it holds that $\int_A A_\tau^\pi(s, a)\pi(da|s) = 0$. Hence taking the logarithm of (4) we have

$$\ln \frac{d\pi^{n+1}}{d\mu}(s, a) - \ln \frac{d\pi^n}{d\mu}(s, a) = -\lambda A_\tau^{\pi^n}(s, a) - \ln \int_A e^{-\lambda A_\tau^{\pi^n}(s, a)} \pi^n(da|s).$$

Interpolating in the time variable and letting $\lambda \to 0$ we expect to retrieve the Fisher–Rao gradient flow for the policies

$$\partial_t \ln \frac{d\pi_t}{d\mu}(s, a) = -\left( A_\tau^{\pi_t}(s, a) - \int_A A_\tau^{\pi_t}(s, a)\pi_t(da|s) \right) = -A_\tau^{\pi_t}(s, a). \tag{5}$$

Note that the soft advantage formally corresponds to the functional derivative of the value function with respect to the policy $\pi^n$ and thus (5) can be seen as a gradient flow of the value function over the space of kernels $\mathcal{P}(A|S)$ (see Kerimkulov et al. (2025a) for a detailed description of the functional derivative).

**Remark 2.1.** *In the case where the advantage function is fully accessible for all $t \geq 0$, Kerimkulov et al. (2025a)[Theorem 2.8] shows that the entropy regularisation in the value function induces an exponential convergence to the optimal policy.*

*More specifically, their result shows that for all $t \geq 0$ we have*

$$0 \leq V_\tau^{\pi_t}(\rho) - V_\tau^{\pi^*}(\rho) \leq \frac{\tau}{(1-\gamma)(e^{\tau t} - 1)} \left( \int_S \mathrm{KL}(\pi^*(\cdot|s)|\pi_0(\cdot|s)) d_\rho^{\pi^*}(ds) \right).$$

*If the action space has finite cardinality and $\pi_0$ is chosen to be uniform we see that $\mathrm{KL}(\pi^*(\cdot|s)|\pi_0(\cdot|s)) \leq \log|A|$ for all $s \in S$, where $|A|$ represents the cardinality of the action*

*space. One can then let $\tau \to 0$ in the above estimate to formally obtain convergence rate of order $1/t$ for the unregularised problem.*

*In the setting of general action spaces $\mathrm{KL}(\pi^*(\cdot|s)|\pi_0(\cdot|s))$ is finite only if the density $\frac{d\pi^*}{d\pi_0}$ exists. However, by the dynamic programming principle for the unregularised problem (Hernández-Lerma & Lasserre, 1996, Theorem 4.2.3) shows that the optimal policies will have support on a mixture of Dirac distributions. Therefore, $\mathrm{KL}(\pi^*(\cdot|s)|\pi_0(\cdot|s))$ will be finite only if $\pi_0$ is also a mixture of Dirac distributions with support which contains the support of $\pi^*(\cdot|s)$ for all $s \in S$. It is not realistic to assume that one can guess the initial policy $\pi_0$ which will have the above property. However, in the entropy regularised case, Theorem A.1 tells us $\pi^*(\cdot|s)$ has full support on $A$ and so one simply has to choose $\pi_0(\cdot|s)$ to have full support on the action space $A$ for all $s \in S$.*

## 3 ACTOR CRITIC METHODS

Given some feature mapping $\phi : S \times A \to \mathbb{R}^N$, we parametrise the state-action value function as $Q(s, a; \theta) := \langle \theta, \phi(s, a) \rangle$. Moreover, we let the approximate soft advantage function be defined as

$$A(s, a; \theta) = Q(s, a; \theta) + \tau \ln \frac{d\pi}{d\mu}(s, a) - \int_A \left( Q(s, a; \theta) + \tau \ln \frac{d\pi}{d\mu}(s, a) \right) \pi(da|s).$$

The Mean Squared Bellman Error (MSBE) is defined as

$$\mathrm{MSBE}(\theta, \pi) = \frac{1}{2} \int_{S \times A} (Q(s, a; \theta) - \mathrm{T}^\pi Q(s, a; \theta))^2 d_\beta^\pi(da, ds)$$

where for some fixed $\beta \in \mathcal{P}(S \times A)$, $d_\beta^\pi \in \mathcal{P}(S \times A)$ is the state-action occupancy measure defined in (10). Given that $\beta \in \mathcal{P}(S \times A)$ has full support, by (3) it holds that $\mathrm{MSBE}(\theta, \pi) = 0$ if and only if $Q(s, a; \theta) = Q_\tau^\pi(s, a)$ for all $s \in S$ and $a \in A$. Hence one approach to implementing the policy mirror descent updates is to calculate the optimal parameters for $Q(s, a; \theta)$ by minimising the MSBE at each policy mirror descent iteration and then update the policy using variable steps $\{\lambda_n\}_{n \geq 0}$. This reads as

$$\begin{cases} \theta^{n+1} = \underset{\theta \in \mathbb{R}^N}{\arg\min}\, \mathrm{MSBE}(\theta, \pi^n), \\ \dfrac{d\pi^{n+1}}{d\pi^n}(a, s) = \dfrac{\exp\left(-\lambda_n A(s, a; \theta^{n+1})\right)}{\int_A \exp\left(-\lambda_n A(s, a; \theta^{n+1})\right) \pi^n(da|s)}. \end{cases} \tag{6}$$

To avoid fully solving the $\arg\min$ in (6) for each policy update, one can update the critic using a semi-gradient descent on a different timescale to the policy update. Let $\{h_n\}_{n \geq 0}$ be the step-sizes of the critic at iteration $n \geq 0$. Let the semi-gradient $g : \mathbb{R}^N \times \mathcal{P}(A|S) \to \mathbb{R}^N$ of the MSBE with respect to $\theta$ be

$$g(\theta, \pi) := \int_{S \times A} (Q(s, a; \theta) - \mathrm{T}^\pi Q(s, a; \theta))\phi(s, a) d_\beta^\pi(da, ds). \tag{7}$$

The full $\arg\min$ update in (6) is then replaced by

$$\theta^{n+1} = \theta^n - h_n g(\theta^n, \pi^n),$$

where timescale separation $\eta_n := \frac{h_n}{\lambda_n} > 1$ ensures that the critic is updated on a much faster timescale than the policy to improve the local estimation of the policy updates.

With general action spaces, which allow the KL term may be unbounded, one may need to go even further and choose a scheme which does several (and possibly increasing) number of updates of the critic before doing an actor update.

In this paper we focus on a continuous-time idealisation of the above which is presented in the next section.

## 4 DYNAMICS

We study the stability and convergence of the two-timescale actor-critic Mirror Descent scheme in the continuous-time limit. Let $\eta : [0, \infty) \to [1, \infty)$ be a non-decreasing function representing the timescale separation, then for some $\theta_0 \in \mathbb{R}^N$, $\pi_0 \in \Pi_\mu$ and $\beta \in \mathcal{P}(S \times A)$, we have the following coupled dynamics

$$\frac{d\theta_t}{dt} = -\eta_t g(\theta_t, \pi_t), \quad \theta_0 = \theta^0 \in \mathbb{R}^N, \tag{8}$$

$$\partial_t \pi_t(da|s) = -A(s, a; \theta_t) \pi_t(da|s), \quad t \geq 0, \quad \pi_0 = \pi^0 \in \Pi_\mu, \tag{9}$$

where $g : \mathbb{R}^N \times \mathcal{P}(A|S)$ is the semi-gradient of the MSBE defined in (7). We refer to (9) as the approximate Fisher–Rao Gradient flow.

We perform our analysis under the following assumptions.

**Assumption 4.1** ($Q_\tau^\pi$-realisability). *For all $\pi \in \Pi_\mu$ there exists $\theta_\pi \in \mathbb{R}^N$ such that $Q^\pi(s, a) = \langle \theta_\pi, \phi(s, a) \rangle$ for all $(s, a) \in S \times A$.*

A simple example of when this holds is in the tabular case, where one can choose $\phi$ to be a one-hot encoding of the state-action space. Moreover, all linear MDPs are $Q^\pi$-realisable. In a linear MDP there exists exists $\phi : S \times A \to \mathbb{R}^N$, $w \in \mathbb{R}^N$ and a sequence $\{\psi_i\}_{i=1}^N$ with $\psi_i \in \mathcal{M}(S)$ such that for all $(s, a) \in S \times A$,

$$c(s, a) = \langle w, \phi(s, a) \rangle, \qquad P(ds' \mid s, a) = \sum_{i=1}^N \phi_i(s, a) \psi_i(ds').$$

In this case it holds that $(\theta_\pi)_i = w_i + \int_S V^\pi(s') \psi_i(ds')$. Assumption 4.1 can be seen as a convention to omit function approximation errors in the final convergence results. This assumption, or the presence of approximation errors in convergence results, are widely present in the actor-critic literature (Cayci et al. (2024a), Zhang et al. (2020), Zanette et al. (2021), Hong et al. (2023),Qiu et al. (2021)).

More recently, Lin et al. (2025) derives some weaker ordering conditions in the bandit case (empty state space) which guarantee the convergence of soft-max policy gradient in the tabular setting beyond realisability. However as of now it is unclear how this applies to MDPs and also fundamentally depends on the finite cardinality of the action space.

Since for all $\pi \in \Pi_\mu$ we know that $Q_\tau^\pi \in B_b(S \times A)$ we also have $Q_\tau^\pi \in L^2(S \times A; \beta)$, which is a Hilbert space. By (Brezis, 2011, Theorem 5.11), Assumption 4.1 holds in the limit $N \to \infty$ when $\phi_i$ are the basis functions of $L^2(S \times A; \beta)$. However, analysis in such a Hilbert space becomes more involved and intricate and is the result of ongoing work. Combining this approach with careful truncation of the basis functions has demonstrated empirical success in Ma et al. (2024); Ren et al. (2023).

**Assumption 4.2.** *For all $(s, a) \in S \times A$ it holds that $|\phi(s, a)| \leq 1$.*

Assumption 4.2 is purely for convention and is without loss of generality in the finite-dimensional case.

**Assumption 4.3.** *Let $\beta \in \mathcal{P}(S \times A)$ be fixed. Then*

$$\lambda_\beta := \lambda_{\min} \left( \int_{S \times A} \phi(s, a) \phi(s, a)^\top \beta(ds\, da) \right) > 0.$$

Note that unlike the analogous assumptions imposed in Hong et al. (2023), Assumption 4.3 is independent of the policy. This property allows us to remove any dependence on the continuity of eigenvalues.

**Definition 4.1.** *For all $\pi \in \Pi_\mu$ and $\zeta \in \mathcal{P}(S \times A)$, the squared loss with respect to $\zeta$ is defined as*

$$L(\theta, \pi; \zeta) = \frac{1}{2} \int_{S \times A} (\langle \theta, \phi(s, a) \rangle - Q_\tau^\pi(s, a))^2 \zeta(da, ds)$$

*where $Q_\tau^\pi$ is defined in (2).*

A straightforward calculation given in Lemma B.3 shows that due to Lemma A.1 and Assumption 4.3, for any $\pi \in \Pi_\mu$ it holds that $L(\cdot, \pi; d_\beta^\pi)$ is $(1-\gamma)\lambda_\beta$-strongly convex.

The following result then connects the geometry of the semi-gradient of the MSBE and the gradient of $L(\cdot, \pi; \beta)$, which can be seen as an extension of Lemma 3 of Bhandari et al. (2021) to the current entropy regularised setting.

**Lemma 4.1.** *Let Assumption 4.1 hold. Then for all $\theta \in \mathbb{R}^N$ and $\pi \in \Pi_\mu$ it holds that*

$$-\langle g(\theta, \pi), \theta - \theta_\pi \rangle \leq -(1-\sqrt{\gamma})(1-\gamma)\langle \nabla_\theta L(\theta, \pi; \beta), \theta - \theta_\pi \rangle$$

*with*

$$\nabla_\theta L(\theta, \pi; \beta) = \int_{S \times A} (\langle \theta, \phi(s,a)\rangle - Q_\tau^\pi(s,a))\phi(s,a)\beta(da, ds).$$

See Appendix B.1 for a proof.

## 5 STABILITY

In this section we analyse the stability of the coupled actor-critic flow. Let $(\theta_t, \pi_t)_{t \geq 0}$ be given by the system (8)-(9). Under mild assumptions, Corollary 5.1 shows that for all $s \in S$, $\overline{\mathrm{KL}}(\pi_t(\cdot|s)|\mu)$ does not blow up in finite time in the sense that there in no $T > 0$ and no $s \in S$ such that $\lim_{t \nearrow T} \mathrm{KL}(\pi_t(\cdot|s)|\mu) = +\infty$. Existence of such a time $T > 0$ would result in a singularity in the actor-critic dynamics.

Throughout this section, to ease notation we let

$$\Gamma := \lambda_\beta(1-\gamma)(1-\sqrt{\gamma}), \quad \mathrm{K}_t := \sup_{s \in S} \mathrm{KL}(\pi_t(\cdot|s)|\mu),$$

with $\lambda_\beta > 0$ the constant from Assumption 4.3.

Using Lemma A.1, Lemma 5.1 then establishes the effect of the coupling and timescale separation in the actor-critic flow and its effect on the stability of the critic parameters.

**Lemma 5.1.** *Let Assumptions 4.2 and 4.3 hold. Then for all $t \geq 0$ it holds that*

$$\frac{1}{2\eta_t}\frac{d}{dt}|\theta_t|^2 \leq -\frac{\Gamma}{2}|\theta_t|^2 + \frac{\tau^2\gamma^2\mathrm{K}_t^2}{\Gamma} + \frac{|c|_{B_b(S \times A)}^2}{\Gamma}$$

See Appendix C.1 for a proof. By connecting the result from Lemma 5.1 with the approximate Fisher–Rao gradient flow, we are able to establish a Grönwall-type inequality for the KL divergence of the policies with respect to the reference measure. Lemma 5.1 also illustrates that the coupled actor-critic flow is a forcing-damping system, where the damping comes from the strong convexity of the loss $\theta \mapsto L(\theta, \pi_t; d_\beta^{\pi_t})$ and the forcing coming from the policy updates manifesting as the $\mathrm{K}_t$ term on the right-hand-side of the estimate. Here we can see that in the finite action space setting, the forcing $\mathrm{K}_t$ term is upper bounded by a constant and thus we can arrive at stability straight away. In the current setting this is not possible and we must perform analysis over $\mathcal{P}(A|S)$ on the approximate Fisher–Rao gradient flow to arrive at stability.

**Theorem 5.1.** *Let Assumptions 4.2 and 4.3 hold. Let $\eta_0 > \frac{\tau}{\Gamma}$. Then there exists constants*

$$a_1 = a_1\left(\tau, \eta_0, \gamma, \lambda_\beta, |c|_{B_b(S \times A)}, \left|\frac{d\pi_0}{d\mu}\right|_{B_b(S \times A)}\right) > 0$$

*and $a_2 = a_2(\tau, \eta_0, \gamma, \lambda_\beta) > 0$ such that for all $\gamma \in (0,1)$ and $t \geq 0$ it holds that*

$$\mathrm{K}_t^2 \leq a_1 + a_2 \int_0^t e^{-\tau(t-r)}\mathrm{K}_r^2 \, dr.$$

See Appendix C.2 for a proof. Through applications of Grönwall's Lemma (Lemma A.3), two direct corollaries of Theorem 5.1 show that the KL divergence of the policies with respect to the reference measure and the critic parameters do not blow up in finite time.

**Corollary 5.1** (Stability of $\pi_t$). *Under the same assumptions as Theorem 5.1, for all $\gamma \in (0,1)$, $s \in S$ and $t \geq 0$ it holds that*

$$\mathrm{KL}(\pi_t(\cdot|s)|\mu)^2 \leq a_1 e^{a_2 t}.$$

**Corollary 5.2** (Stability of $\theta_t$). *Under the same assumptions as Theorem 5.1, suppose that there exists $\alpha > 0$ such that $\frac{d}{dt}\eta_t \leq \alpha\eta_t$. Then for all $\gamma \in (0,1)$ there exists $r_1, r_2 > 0$ such that for all $t \geq 0$ it holds that*

$$|\theta_t| \leq r_1 e^{r_2 t}.$$

See Appendix C.3 and C.4 for the proofs.

If the MDP has sufficiently small effective time horizon due to a sufficiently small discounting factor and thus is in a sense regularised, the KL divergence of the policies with respect to the reference measure remains uniformly bounded along the flow, see Corollaries E.1 and E.2.

# 6 CONVERGENCE

In this section we will present final three key components before we get to the final convergence result for the coupled actor-critic flow. First, we characterise the time derivative of the state-action value function along the approximate gradient flow for the policies.

**Lemma 6.1.** *For all $t \geq 0$ and $(s,a) \in S \times A$, it holds that*

$$\frac{d}{dt}Q_\tau^{\pi_t}(s,a) = \frac{\gamma}{1-\gamma}\int_S \left(\int_{S \times A} A_\tau^{\pi_t}(s'',a'')\partial_t\pi_t(da''|s'')d^{\pi_t}(ds''|s')\right)P(ds'|s,a)$$

See Appendix D.1 for a proof. Observe that in the exact setting, where $\partial_t\pi_t = -A_\tau^{\pi_t}$ as in (5), we obtain the dissipative property of $\{Q_\tau^{\pi_t}\}_{t\geq 0}$ along the flow

$$\frac{d}{dt}Q_\tau^{\pi_t}(s,a) = \frac{-\gamma}{1-\gamma}\int_S \left(\int_{S \times A} A_\tau^{\pi_t}(s'',a'')^2 d^{\pi_t}(ds''|s')\right)P(ds'|s,a) \leq 0.$$

Second, Theorem 6.1 shows that the actor-critic flow maintains the exponential convergence to the optimal policy induced by the $\tau$-regularisation up to a error term arising from not solving the critic to full accuracy.

**Theorem 6.1.** *Let $\{\pi_t, \theta_t\}_{t\geq 0}$ be the trajectories of the actor critic flow. Let Assumptions 4.1 and 4.2 hold. Then for all $t > 0$ it holds that*

$$\min_{r \in [0,t]} V_\tau^{\pi_r}(\rho) - V_\tau^{\pi^*}(\rho) \leq \frac{\tau}{2(1-\gamma)(1-e^{-\frac{\tau}{2}t})}\left(e^{-\frac{\tau}{2}t}\int_S \mathrm{KL}(\pi^*(\cdot|s)|\pi_0(\cdot|s))d_\rho^{\pi^*}(ds)\right.$$
$$\left. + \frac{1}{2\tau}\int_0^t e^{-\frac{\tau}{2}(t-r)}|\theta_r - \theta_{\pi_r}|^2 dr\right)$$

See Appendix D.2 for a proof. Theorem 6.1 shows that the exponentially weighted error term determines the rate of convergence of the actor-critic dynamics.

Third, Theorem 6.2 shows that this error term decays exponentially up to an integral which now depends on the rate of change of the true state-action value function and the timescale separation.

**Theorem 6.2.** *Let Assumptions 4.1, 4.2 and 4.3 hold. Let $\eta_0 > \frac{1}{\Gamma}$ and $0 < \tau < 1$. Then for all $t \geq 0$ there exists constants $b_1, b_2 > 0$ such that*

$$\int_0^t e^{-\frac{\tau}{2}(t-r)}|\theta_r - \theta_{\pi_r}|^2 dr \leq b_1 e^{-\frac{\tau}{2}t} + b_2 \int_0^t e^{-\frac{\tau}{2}(t-r)}\frac{1}{\eta_r}\left|\frac{d}{dr}\theta_{\pi_r}\right|^2 dr.$$

See Appendix D.3 for a proof.

Finally we are ready to present the main result of the paper. Using Corollary 5.1 and by choosing $\eta_t$ such that the critic flows runs much faster than the actor, Theorem 6.3 below demonstrates an exponential convergence to the optimal policy for all $\gamma \in (0,1)$.

**Theorem 6.3.** *Under the same assumptions as Theorem 6.2, there exists $k_1 > 0$ with $\eta_t = \eta_0 e^{k_1 t}$ and $k_2 > 0$ such that for all $\gamma \in (0, 1)$ and $t > 0$ it holds that*

$$\min_{r \in [0,t]} V_\tau^{\pi_r}(\rho) - V_\tau^{\pi^*}(\rho) \leq \frac{\tau e^{-\frac{\tau}{2}t}}{2(1-\gamma)(1 - e^{-\frac{\tau}{2}t})} \left( \int_S \mathrm{KL}(\pi^*(\cdot|s)|\pi_0(\cdot|s)) d_\rho^{\pi^*}(ds) + \frac{k_2}{2\tau} \right)$$

See Appendix D.4 for a proof.

Corollary E.3 then shows that if the MDP is sufficiently regularised through a small discounting factor, one can arrive at convergence for a much more general class of timescale separation functions $t \mapsto \eta_t$.

## 7 LIMITATIONS

In this work, we only study the continuous-time dynamics of the actor-critic algorithm. Although this formulation gives insights into the discrete counterpart, a rigorous treatment of the discrete-time setting is more realistic for practical purposes and is left for future research.

Moreover, for the purposes of analysis our critic approximation is linear while in practice non-linear neural networks are used to approximate the critic.

Finally, our work assumes all integrals are evaluated exactly, in particular the semi-gradient (7). In practice these would need to be estimated from samples leading to additional Monte-Carlo errors. To fully analyse this is left for future work.

## 8 ACKNOWLEDGEMENTS

DZ was supported by the EPSRC Centre for Doctoral Training in Mathematical Modelling, Analysis and Computation (MAC-MIGS) funded by the UK Engineering and Physical Sciences Research Council (grant EP/S023291/1), Heriot-Watt University and the University of Edinburgh. The work on this project by DŠ was partially supported by a grant from the Simons Foundation. DŠ and LS acknowledge funding from the UKRI Prosperity Partnerships grant APP43592: AI$^2$ - Assurance and Insurance for Artificial Intelligence, which supported this work. The authors would like to thank the Isaac Newton Institute for Mathematical Sciences, Cambridge, for support and hospitality during the programme "Bridging Stochastic Control And Reinforcement Learning", where work on this paper was undertaken. This work was supported by EPSRC grant EP/V521929/1.

Last but not least, the authors would like to thank the anonymous reviewers whose insightful comments have helped improve the paper.

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

## A    KNOWN PROPERTIES MDPS AND OTHER USEFUL RESULTS

The state-occupancy kernel $d^\pi \in \mathcal{P}(S|S)$ is defined by

$$d^\pi(ds'|s) = (1-\gamma)\sum_{n=0}^{\infty}\gamma^n P_\pi^n(ds'|s),$$

where $P_\pi^n$ is the $n$-times product of the kernel $P_\pi$ with $P_\pi^0(ds'|s) := \delta_s(ds')$. Moreover, for each $\pi \in \mathcal{P}(A|S)$ and $(s,a) \in S \times A$, we define the state-action occupancy kernel as

$$d^\pi(ds, da|s, a) = (1-\gamma)\sum_{n=0}^{\infty}\gamma^n (P^\pi)^n(ds, da|s, a)$$

where $(P^\pi)^n$ is the $n$-times product of the kernel $P^\pi$ with $(P^\pi)^0(ds', da'|s, a) := \delta_{(s,a)}(ds', da')$. Given some initial state-action distribution $\beta \in \mathcal{P}(S \times A)$ with initial state distribution given by $\rho(ds) = \int_A \beta(da, ds)$, we define the state-occupancy and state-action occupancy measures as

$$d_\rho^\pi(ds) = \int_S d^\pi(ds|s')\rho(ds'), \quad d_\beta^\pi(ds, da) = \int_{S \times A} d^\pi(ds, da|s', a')\beta(da', ds'). \tag{10}$$

Note that for all $E \in \mathcal{B}(S \times A)$, by defining the linear operator $J_\pi : \mathcal{P}(S \times A) \to \mathcal{P}(S \times A)$ as

$$J_\pi\beta(E) = \int_{S \times A} P^\pi(E|s', a')\beta(ds', da'), \tag{11}$$

it directly holds that

$$d_\beta^\pi(da, ds) = (1-\gamma)\sum_{n=0}^{\infty}\gamma^n J_\pi^n\beta(da, ds),$$

with $J_\pi^n$ the $n$-fold product of the operator $J_\pi$ with $J_\pi^0 = I$, the identity operator on $\mathcal{P}(S \times A)$. The following lemma establishes properties of the state-action occupancy measure defined in (10) and which are useful in the proofs.

**Lemma A.1.** *For all $\pi \in \mathcal{P}(A|S)$, $\beta \in \mathcal{P}(S \times A)$ and $E \in \mathcal{B}(S \times A)$ it holds that*

$$d_{J^\pi\beta}^\pi(E) = J^\pi d_\beta^\pi(E).$$

*Moreover, for all $\gamma \in (0, 1)$ we have*

$$d_\beta^\pi(E) - \gamma d_{J^\pi\beta}^\pi(E) = (1-\gamma)\beta(E). \tag{12}$$

*Proof.* For any $\beta \in \mathcal{P}(S \times A)$, $\pi \in \mathcal{P}(A|S)$ and $E \in \mathcal{B}(S \times A)$, it holds that

$$d_{J_\pi\beta}^\pi(E) = (1-\gamma)\sum_{n=0}^{\infty}\gamma^n(J_\pi^n J_\pi\beta)(E)$$

$$= J_\pi d_\beta^\pi(E)$$

where we just used the associativity of the operator $J_\pi$. Furthermore by letting $m = n + 1$ it holds that

$$d_{J_\pi\beta}^\pi(E) = (1-\gamma)\sum_{n=0}^{\infty}\gamma^n J_\pi^{n+1}\beta(E)$$

$$= (1-\gamma)\sum_{m=1}^{\infty}\gamma^{m-1}J_\pi^m\beta(E)$$

$$= \frac{1-\gamma}{\gamma}\sum_{m=1}^{\infty}\gamma^m J_\pi^m\beta(E)$$

$$= \frac{1}{\gamma}(d_\beta^\pi(E) - (1-\gamma)\beta(E)).$$

Rearranging concludes the proof. □

**Theorem A.1** (Dynamic Programming Principle). *Let $\tau > 0$. The optimal value function $V_\tau^*$ is the unique bounded solution of the following Bellman equation:*

$$V_\tau^*(s) = -\tau \ln \int_A \exp\left(-\frac{1}{\tau} Q_\tau^*(s, a)\right) \mu(da),$$

*where $Q_\tau^* \in B_b(S \times A)$ is defined by*

$$Q_\tau^*(s, a) = c(s, a) + \gamma \int_S V_\tau^*(s') P(ds'|s, a), \quad \forall (s, a) \in S \times A.$$

*Moreover, there is an optimal policy $\pi^* \in \mathcal{P}(A|S)$ given by*

$$\pi^*(da|s) = \exp\left(-\frac{1}{\tau}(Q_\tau^*(s, a) - V_\tau^*(s))\right) \mu(da), \quad \forall s \in S.$$

*Finally, the value function $V_\tau^\pi$ is the unique bounded solution of the following Bellman equation for all $s \in S$*

$$V_\tau^\pi(s) = \int_A \left(Q_\tau^\pi(s, a) + \tau \ln \frac{d\pi}{d\mu}(a, s)\right) \pi(da|s).$$

The performance difference lemma, first introduced for tabular unregularised MDPs, has become fundamental in the analysis of MDPs as it acts a substitute for the strong convexity of the $\pi \mapsto V_\tau^\pi$ if the state-occupancy measure $d_\rho^\pi$ is ignored (e.g Kakade & Langford (2002), Zhang et al. (2021), Ju & Lan (2024)). By virtue of Kerimkulov et al. (2025a), we have the following performance difference for entropy regularised MDPs in Polish state and action spaces.

**Lemma A.2** (Performance difference). *For all $\rho \in \mathcal{P}(S)$ and $\pi, \pi' \in \Pi_\mu$,*

$$V_\tau^\pi(\rho) - V_\tau^{\pi'}(\rho)$$
$$= \frac{1}{1 - \gamma} \int_S \left[\int_A \left(Q_\tau^{\pi'}(s, a) + \tau \ln \frac{d\pi'}{d\mu}(a, s)\right)(\pi - \pi')(da|s) + \tau \operatorname{KL}(\pi(\cdot|s)|\pi'(\cdot|s))\right] d_\rho^\pi(ds).$$

**Lemma A.3** (Grönwall). *Let $\lambda(s) \geq 0$, $a = a(s)$, $b = b(s)$ and $y = y(s)$ be locally integrable, real-valued functions defined on $[0, T]$ such that $y$ is also locally integrable and for almost all $s \in [0, T]$,*

$$y(s) + a(s) \leq b(s) + \int_0^s \lambda(t) y(t) dt.$$

*Then*

$$y(s) + a(s) \leq b(s) + \int_0^s \lambda(t) \left[\int_0^t \lambda(r)(b(r) - a(r)) dr\right] dt, \quad \forall s \in [0, T].$$

*Furthermore, if $b$ is monotone increasing and $a$ is non-negative, then*

$$y(s) + a(s) \leq b(s) e^{\int_0^s \lambda(r) dr}, \quad \forall s \in [0, T].$$

## B  Auxiliary results

**Lemma B.1.** *For some $\beta \in \mathcal{P}(S \times A)$, let $d_\beta^\pi \in \mathcal{P}(S \times A)$ be the state-action occupancy measure. Moreover let $\kappa(ds, da, ds', da') := P^\pi(ds', da'|s, a) d_\beta^\pi(ds, da)$. Then for any $\pi \in \Pi_\mu$ and any integrable $f : S \times A \to \mathbb{R}$, it holds that*

$$\int_{S \times A \times S \times A} f(s, a) f(s', a') \kappa(ds, da, ds', da') \leq \frac{1}{\sqrt{\gamma}} \int_{S \times A} f(s, a)^2 d_\beta^\pi(ds, da)$$

*Proof.* By Hölder's inequality, it holds that

$$\int_{S \times A \times S \times A} f(s, a) f(s', a') \kappa(ds, da, ds', da') \tag{13}$$
$$\leq \left(\int_{S \times A \times S \times A} f(s, a)^2 \kappa(ds, da, ds', da')\right)^{\frac{1}{2}} \left(\int_{S \times A \times S \times A} f(s', a')^2 \kappa(ds, da, ds', da')\right)^{\frac{1}{2}}.$$

Moreover, observe that

$$\int_{S \times A \times S \times A} f(s,a)^2 \kappa(ds, da, ds', da') = \int_{S \times A} \left( \int_{S \times A} P^\pi(ds', da'|s,a) \right) f(s,a)^2 d_\beta^\pi(ds, da)$$
$$= \int_{S \times A} f(s,a)^2 d_\beta^\pi(ds, da),$$

hence (13) becomes

$$\left( \int_{S \times A \times S \times A} f(s,a)^2 \kappa(ds, da, ds', da') \right)^{\frac{1}{2}} \left( \int_{S \times A \times S \times A} f(s',a')^2 \kappa(ds, da, ds', da') \right)^{\frac{1}{2}}$$
$$\leq \left( \int_{S \times A} f(s,a)^2 d_\beta^\pi(ds, da) \right)^{\frac{1}{2}} \left( \int_{S \times A \times S \times A} f(s',a')^2 \kappa(ds, da, ds', da') \right)^{\frac{1}{2}}.$$

Now by the first part of Lemma A.1, it holds that

$$\int_{S \times A \times S \times A} f(s',a')^2 \kappa(ds, da, ds', da') = \int_{S \times A \times S \times A} f(s',a')^2 P^\pi(ds', da'|s,a) d_\beta^\pi(ds, da)$$
$$= \int_{S \times A} f(s,a)^2 d_{J^\pi \beta}^\pi(ds, da),$$

where $J^\pi : \mathcal{P}(S \times A) \to \mathcal{P}(S \times A)$ is defined in (11). Then by the second part of Lemma A.1 we have

$$\left( \int_{S \times A} f(s,a)^2 d_\beta^\pi(ds, da) \right)^{\frac{1}{2}} \left( \int_{S \times A \times S \times A} f(s',a')^2 \kappa(ds, da, ds', da') \right)^{\frac{1}{2}}$$
$$\leq \left( \int_{S \times A} f(s,a)^2 d_\beta^\pi(ds, da) \right)^{\frac{1}{2}} \left( \int_{S \times A} f(s,a)^2 d_{J^\pi \beta}^\pi(ds, da) \right)^{\frac{1}{2}}$$
$$\leq \frac{1}{\sqrt{\gamma}} \int_{S \times A} f(s,a)^2 d_\beta^\pi(ds, da),$$

which concludes the proof. $\square$

To alleviate notation let $Q_t(s,a) := Q(s,a;\theta_t)$ and $A_t(s,a) := A(s,a;\theta_t)$.

**Lemma B.2.** *For some $\theta_0 \in \mathbb{R}^N$ and $\pi_0 \in \Pi_\mu$, let $\{\pi_t, \theta_t\}_{t \geq 0}$ be the trajectory of coupled actor-critic flow. Moreover let $K_t = \sup_{s \in S} \mathrm{KL}(\pi_t(\cdot|s)|\mu)$. There exists $C_1 > 0$ such that for all $t \geq 0$ it holds that*

$$\sup_{s \in S} |\partial_t \pi_t(\cdot|s)|_{\mathcal{M}(A)} \leq |A_t|_{B_b(S \times A)},$$

$$|A_t|_{B_b(S \times A)} \leq 2 |Q_t|_{B_b(S \times A)} + 2\tau \left| \ln \frac{d\pi_t}{d\mu} \right|_{B_b(S \times A)},$$

$$|Q_\tau^{\pi_t}|_{B_b(S \times A)} \leq \frac{1}{1 - \gamma} \left( |c|_{B_b(S \times A)} + \tau \gamma K_t \right),$$

$$\left| \ln \frac{d\pi_t}{d\mu} \right|_{B_b(S \times A)} \leq C_1 + \frac{2}{\tau} \sup_{r \in [0,t]} |\theta_r| + \sup_{r \in [0,t]} K_r.$$

*Proof.* The first claim $\sup_{s \in S} |\partial_t \pi_t(\cdot|s)|_{\mathcal{M}(A)} \leq |A_\tau^{\pi_t}|_{B_b(S \times A)}$ follows trivially from the definition of the approximate Fisher–Rao gradient flow defined in (9). Moreover, it holds that

$$|A_t|_{B_b(S \times A)} = \left| Q_t + \tau \ln \frac{d\pi_t}{d\mu} - \int_A \left( Q_t(\cdot, a) + \tau \ln \frac{d\pi_t}{d\mu}(\cdot, a) \right) \pi_t(da|\cdot) \right|_{B_b(S \times A)}$$
$$\leq 2 \left| Q_t + \tau \ln \frac{d\pi_t}{d\mu} \right|_{B_b(S \times A)}$$
$$\leq 2 |Q_t|_{B_b(S \times A)} + 2\tau \left| \ln \frac{d\pi_t}{d\mu} \right|_{B_b(S \times A)}$$

where we used the triangle inequality in the final inequality. Moreover, the state-action value function $Q_\tau^{\pi_t}$ is a fixed point of the Bellman operator defined in (3). Hence, for all $(s, a) \in S \times A$, we have

$$Q_\tau^{\pi_t}(s, a) = c(s, a) + \gamma \int_{S \times A} Q_\tau^{\pi_t}(s', a') \, P^{\pi_t}(ds', da'|s, a) + \tau\gamma \int_S \text{KL}(\pi_t(\cdot|s')\|\mu) \, P(ds'|s, a).$$

Taking absolute values and using the triangle inequality we have

$$|Q_\tau^{\pi_t}(s, a)| \leq |c|_{B_b(S \times A)} + \gamma |Q_\tau^{\pi_t}|_{B_b(S \times A)} + \tau\gamma \sup_{s' \in S} \text{KL}(\pi_t(\cdot|s')\|\mu)$$

$$= |c|_{B_b(S \times A)} + \gamma |Q_\tau^{\pi_t}|_{B_b(S \times A)} + \tau\gamma \text{K}_t.$$

Taking the supremum over $(s, a) \in S \times A$ on the left-hand side yields

$$|Q_\tau^{\pi_t}|_{B_b(S \times A)} \leq |c|_{B_b(S \times A)} + \gamma |Q_\tau^{\pi_t}|_{B_b(S \times A)} + \tau\gamma \text{K}_t.$$

Rearranging gives

$$(1 - \gamma) |Q_\tau^{\pi_t}|_{B_b(S \times A)} \leq |c|_{B_b(S \times A)} + \tau\gamma \text{K}_t,$$

which is the desired bound. Recall the approximate Fisher–Rao gradient flow for the policies $\{\pi_t\}_{t \geq 0}$, which for all $t \geq 0$ and for all $(s, a) \in S \times A$ is given by

$$\partial_t \ln \frac{d\pi_t}{d\mu}(s, a) = -\left( Q_t(s, a) + \tau \ln \frac{d\pi_t}{d\mu}(a, s) - \int_A \left( Q_t(s, a') + \tau \ln \frac{d\pi_t}{d\mu}(a', s) \right) \pi_t(da'|s) \right).$$

Duhamel's principle yields for all $t \geq 0$ that

$$\ln \frac{d\pi_t}{d\mu}(s, a) = e^{-\tau t} \ln \frac{d\pi_0}{d\mu}(a, s) + \int_0^t e^{-\tau(t-r)} \left( \int_A Q_r(s, a')\pi_r(da'|s) - Q_r(s, a) \right) dr \quad (14)$$

$$+ \tau \int_0^t e^{-\tau(t-r)} \text{KL}(\pi_r(\cdot|s)|\mu) dr.$$

Since $\pi_0 \in \Pi_\mu$, there exists $C_1 \geq 1$ such that $\left| \ln \frac{d\pi_0}{d\mu} \right|_{B_b(S \times A)} \leq C_1$. Then by Assumption 4.2 we have that for all $t \geq 0$,

$$\left| \ln \frac{d\pi_t}{d\mu}(s, a) \right| \leq C_1 + \int_0^t e^{-\tau(t-r)} \left| \int_A Q_r(s, a')\pi_r(da'|s) - Q_r(s, a) \right| dr$$

$$+ \tau \int_0^t e^{-\tau(t-r)} \text{KL}(\pi_r(\cdot|s)\|\mu) \, dr$$

$$\leq C_1 + 2 \int_0^t e^{-\tau(t-r)} |\theta_r| \, dr + \tau \int_0^t e^{-\tau(t-r)} \text{K}_r \, dr$$

$$\leq C_1 + \frac{2}{\tau} \sup_{r \in [0,t]} |\theta_r| + \sup_{r \in [0,t]} \text{K}_r,$$

where in the last inequality we used $\int_0^t e^{-\tau(t-r)} dr \leq \frac{1}{\tau}$. Taking the supremum over $(s, a) \in S \times A$ yields

$$\left| \ln \frac{d\pi_t}{d\mu} \right|_{B_b(S \times A)} \leq C_1 + \frac{2}{\tau} \sup_{r \in [0,t]} |\theta_r| + \sup_{r \in [0,t]} \text{K}_r,$$

which is the desired bound. $\qquad \square$

**Lemma B.3.** *Let Assumption 4.3 hold. Then for all $\pi \in \Pi_\mu$, it holds that $L(\cdot, \pi; d_\beta^\pi)$ is $\lambda_\beta(1 - \gamma)$-strongly convex.*

*Proof.* For any $\xi \in \mathcal{P}(S \times A)$, let $\Sigma_\xi := \int_{S \times A} \phi(s, a)\phi(s, a)^\top \xi(ds, da) \in \mathbb{R}^{N \times N}$. Then by Lemma A.1 and Assumption 4.3 it holds that $\Sigma_{d_\beta^\pi} \succeq (1 - \gamma)\Sigma_\beta \succeq (1 - \gamma)\lambda_\beta I$ and thus $L(\cdot, \pi; d_\beta^\pi)$ is $\lambda_\beta(1 - \gamma)$-strongly convex. $\qquad \square$

## B.1 PROOF OF LEMMA 4.1

*Proof.* Recall that $Q(s, a) = \langle \theta, \phi(s, a) \rangle$ for some $\theta \in \mathbb{R}^N$ and that for all $\pi \in \Pi_\mu$, there exists $\theta_\pi \in \mathbb{R}^N$ such that $Q^\pi(s, a) = \langle \theta_\pi, \phi(s, a) \rangle$ by Assumption 4.1. Then by definition of the semi-gradient of the MSBE $g : \mathbb{R}^N \times \mathcal{P}(A|S) \to \mathbb{R}^N$ in (7), it holds that

$$
\langle g(\theta, \pi), \theta - \theta_\pi \rangle = \left\langle \int_{S \times A} (Q(s, a) - \mathrm{T}^\pi Q(s, a)) \, \phi(s, a) d_\beta^\pi(da, ds), \theta - \theta_\pi \right\rangle
$$

$$
= \left\langle \int_{S \times A} (Q(s, a) - Q_\tau^\pi(s, a)) \phi(s, a) d_\beta^\pi(da, ds), \theta - \theta_\pi \right\rangle
$$

$$
+ \left\langle \int_{S \times A} (Q_\tau^\pi(s, a) - \mathrm{T}^\pi Q(s, a) \phi(s, a) d_\beta^\pi(da, ds), \theta - \theta_\pi \right\rangle
$$

$$
= \left\langle \int_{S \times A} (Q(s, a) - Q_\tau^\pi(s, a)) \phi(s, a) d_\beta^\pi(da, ds), \theta - \theta_\pi \right\rangle
$$

$$
- \gamma \left\langle \int_{S \times A \times S \times A} (Q(s', a') - Q_\tau^\pi(s', a')) \phi(s, a) P^\pi(ds', da'|s, a) d_\beta^\pi(ds, da), \theta - \theta_\pi \right\rangle,
$$

where we added and subtracted the true state-action value function $Q_\tau^\pi \in B_b(S \times A)$ in the second equality and used the fact that it is a fixed point of the Bellman operator defined in (3). To ease notation, let $\varepsilon(s, a) := Q(s, a) - Q_\tau^\pi(s, a)$. Multiplying both sides by $-1$ and using the associativity of the inner product, we have

$$
- \langle g(\theta, \pi), \theta - \theta_\pi \rangle
$$

$$
= - \left\langle \int_{S \times A} \varepsilon(s, a) \phi(s, a) d_\beta^\pi(da, ds), \theta - \theta_\pi \right\rangle
$$

$$
+ \gamma \left\langle \int_{S \times A} \varepsilon(s', a') \phi(s, a) P^\pi(ds', da'|s, a) d_\beta^\pi(ds, da), \theta - \theta_\pi \right\rangle
$$

$$
= - \int_{S \times A} \varepsilon(s, a) \langle \phi(s, a), \theta - \theta_\pi \rangle d_\beta^\pi(da, ds)
$$

$$
+ \gamma \int_{S \times A} \varepsilon(s', a') \langle \phi(s, a), \theta - \theta_\pi \rangle P^\pi(ds', da'|s, a) d_\beta^\pi(ds, da)
$$

$$
= - \int_{S \times A} \varepsilon(s, a)^2 d_\beta^\pi(da, ds)
$$

$$
+ \gamma \int_{S \times A \times S \times A} \varepsilon(s, a) \varepsilon(s', a') P^\pi(ds', da'|s, a) d_\beta^\pi(ds, da)
$$

$$
= I^{(1)} + \gamma I^{(2)}.
$$

Now applying Lemma B.1 to $I^{(2)}$ we have

$$
I^{(2)} := \int_{S \times A \times S \times A} \varepsilon(s, a) \varepsilon(s', a') P^\pi(ds', da'|s, a) d_\beta^\pi(ds, da)
$$

$$
\leq \frac{1}{\sqrt{\gamma}} \int_{S \times A} \varepsilon(s, a)^2 d_\beta^\pi(ds, da).
$$

Thus it holds that

$$
- \langle g(\theta, \pi), \theta - \theta_\pi \rangle \leq I^{(1)} + \gamma I^{(2)}
$$

$$
\leq -(1 - \sqrt{\gamma}) \int_{S \times A} \epsilon(s, a)^2 d_\beta^\pi(da, ds)
$$

$$
= -(1 - \sqrt{\gamma}) \int_{S \times A} (Q(s, a) - Q_\tau^\pi(s, a))^2 d_\beta^\pi(da, ds)
$$

$$
= -(1 - \sqrt{\gamma}) \langle \nabla_\theta L(\theta, \pi; d_\beta^\pi), \theta - \theta_\pi \rangle,
$$

where the last inequality follows from the Assumption 4.1 and the definition of $Q(s, a) = \langle \theta, \phi(s, a) \rangle$.

$\square$

## C    PROOF OF STABILITY RESULTS

### C.1    PROOF OF LEMMA 5.1

*Proof.* Consider the following equation

$$\frac{1}{2\eta_t}\frac{d}{dt}|\theta_t|^2 = \frac{1}{\eta_t}\left\langle\frac{d}{dt}\theta_t, \theta_t\right\rangle \tag{15}$$

$$= -\langle g(\theta_t, \pi_t), \theta_t\rangle$$

$$= -\left\langle\int_{S\times A}(Q_t(s,a) - T^{\pi_t}Q_t(s,a))\,\phi(s,a)\,d^{\pi_t}_\beta(da,ds), \theta_t\right\rangle$$

$$= -\left\langle\int_{S\times A}Q_t(s,a)\phi(s,a)\,d^{\pi_t}_\beta(da,ds), \theta_t\right\rangle$$

$$+ \left\langle\int_{S\times A}T^{\pi_t}Q_t(s,a)\phi(s,a)\,d^{\pi_t}_\beta(da,ds), \theta_t\right\rangle$$

$$:= -J_t^{(1)} + J_t^{(2)}$$

where we used the $\theta_t$ dynamics from (8) in the second equality and the definition of the semi-gradient in the third equality. For any $\pi \in \Pi_\mu$, let $\Sigma^\pi \in \mathbb{R}^{N\times N}$ be

$$\Sigma^\pi = \int_{S\times A}\phi(s,a)\phi(s,a)^\top d^\pi_\beta(da,ds).$$

Then by definition we have that $Q_t(s,a) = \langle\theta_t, \phi(s,a)\rangle$, hence for $J_t^{(1)}$ we have

$$J_t^{(1)} = \left\langle\int_{S\times A}Q_t(s,a)\phi(s,a)\,d^{\pi_t}_\beta(da,ds), \theta_t\right\rangle$$

$$= \left\langle\int_{S\times A}\langle\theta_t, \phi(s,a)\rangle\,\phi(s,a)d^{\pi_t}_\beta(da,ds), \theta_t\right\rangle$$

$$= \left\langle\theta_t, \left(\int_{S\times A}\phi(s,a)\phi(s,a)^\top d^{\pi_t}_\beta(da,ds)\right)\theta_t\right\rangle$$

$$= \langle\theta_t, \Sigma^{\pi_t}\theta_t\rangle \tag{16}$$

Now dealing with $J_t^{(1)}$, expanding the Bellman operator defined in (3) we have

$$J_t^{(2)} = \left\langle\int_{S\times A}T^{\pi_t}Q_t(s,a)\phi(s,a)\,d^{\pi_t}_\beta(da,ds), \theta_t\right\rangle$$

$$= \left\langle\int_{S\times A}c(s,a)\phi(s,a)d^{\pi_t}_\beta(da,ds), \theta_t\right\rangle$$

$$+ \gamma\left\langle\int_{S\times A}\langle\theta_t, \phi(s',a')\rangle\,\phi(s,a)P^{\pi_t}(ds',da'|s,a)d^{\pi_t}_\beta(da,ds), \theta_t\right\rangle$$

$$+ \tau\gamma\left\langle\int_{S\times A}\left(\int_S \mathrm{KL}(\pi_t(\cdot|s'),\mu)P(ds'|s,a)\phi(s,a)d^{\pi_t}_\beta(da,ds)\right), \theta_t\right\rangle$$

$$\le |c|_{B_b(S\times A)}|\theta_t| + \gamma I_t^{(1)} + \tau\gamma I_t^{(2)}$$

where we defined

$$I_t^{(1)} = \left\langle\int_{S\times A}\langle\theta_t, \phi(s',a')\rangle\,\phi(s,a)P^{\pi_t}(ds',da'|s,a)d^{\pi_t}_\beta(da,ds), \theta_t\right\rangle,$$

$$I_t^{(2)} = \left\langle\int_{S\times A}\left(\int_S \mathrm{KL}(\pi_t(\cdot|s'),\mu)P(ds'|s,a)\phi(s,a)d^\pi_\beta(da,ds)\right), \theta_t\right\rangle.$$

Moreover, to ease notation let

$$\mathrm{K}_t := \sup_{s\in S}\mathrm{KL}(\pi_t(\cdot|s)|\mu)$$

and temporarily let $\kappa_t(ds, da, ds', da') := P^{\pi_t}(ds', da'|s, a)d_\beta^{\pi_t}(da, ds)$. Now focusing on $I_t^{(1)}$, it holds that

$$I_t^{(1)} = \left\langle \iint_{S \times A \times S \times A} \langle \theta_t, \phi(s', a') \rangle \phi(s, a) \kappa_t(da', ds', da, ds), \theta_t \right\rangle$$

$$= \int_{S \times A \times S \times A} \langle \theta_t, \phi(s, a) \rangle \langle \theta_t, \phi(s', a') \rangle \kappa_t(ds', da', ds, da).$$

Now using Lemma B.1 with $f = \langle \theta, \phi(\cdot, \cdot) \rangle$ we have

$$I_t^{(1)} \le \frac{1}{\sqrt{\gamma}} \left( \int_{S \times A} \langle \theta_t, \phi(s, a) \rangle^2 d_\beta^{\pi_t}(ds, da) \right)^{\frac{1}{2}} \left( \int_{S \times A} \langle \theta_t, \phi(s, a) \rangle^2 d_\beta^{\pi_t}(ds, da) \right)^{\frac{1}{2}}$$

$$= \frac{1}{\sqrt{\gamma}} \int_{S \times A} \langle \theta_t, \phi(s, a) \rangle^2 d_\beta^{\pi_t}(ds, da)$$

$$= \frac{1}{\sqrt{\gamma}} \langle \theta_t, \Sigma^{\pi_t} \theta_t \rangle.$$

Thus all together it holds that

$$\gamma I_t^{(1)} \le \sqrt{\gamma} \langle \theta_t, \Sigma^{\pi_t} \theta_t \rangle.$$

Now focusing on $I_t^{(2)}$, we have

$$I_t^{(2)} = \left\langle \int_{S \times A} \left( \int_S \mathrm{KL}(\pi_t(\cdot|s'), \mu) P(ds'|s, a) \right) \phi(s, a) d_\beta^{\pi_t}(da, ds), \theta_t \right\rangle$$

$$\le \mathrm{K}_t \left| \int_{S \times A} \phi(s, a) d_\beta^{\pi_t}(ds, da) \right| |\theta_t|$$

$$\le \mathrm{K}_t |\theta_t|$$

where we used Assumption 4.2 in the final inequality. Hence along with (16), (15) becomes

$$\frac{1}{2\eta_t} \frac{d}{dt} |\theta_t|^2 \le -J_t^{(1)} + J_t^{(2)} \tag{17}$$

$$\le -\langle \theta_t, \Sigma^{\pi_t} \theta_t \rangle + |c|_{B_b(S \times A)} |\theta_t| + \gamma I_t^{(1)} + \tau \gamma I_t^{(2)}$$

$$\le -\langle \theta_t, \Sigma^{\pi_t} \theta_t \rangle + \sqrt{\gamma} \langle \theta_t, \Sigma^{\pi_t} \theta_t \rangle + |c|_{B_b(S \times A)} |\theta_t| + \tau \gamma \mathrm{K}_t |\theta_t|$$

$$= -(1 - \sqrt{\gamma}) \langle \theta_t, \Sigma^{\pi_t} \theta_t \rangle + \left( |c|_{B_b(S \times A)} + \tau \gamma \mathrm{K}_t \right) |\theta_t|.$$

Observe that by (12) and Assumption 4.2, $\Sigma^\pi \in \mathbb{R}^{N \times N}$ is positive definite for all $\pi \in \mathcal{P}(A|S)$, hence it holds that

$$\langle \theta_t, \Sigma^{\pi_t} \theta_t \rangle \ge (1 - \gamma) \lambda_\beta |\theta_t|^2.$$

Therefore (17) becomes

$$\frac{1}{2\eta_t} \frac{d}{dt} |\theta_t|^2 \le -(1 - \sqrt{\gamma})(1 - \gamma) \lambda_\beta |\theta_t|^2 + (|c|_{B_b(S \times A)} + \tau \gamma \mathrm{K}_t) |\theta_t|$$

Let $\Gamma := \lambda_\beta (1 - \gamma)(1 - \sqrt{\gamma})$. By Young's inequality, there exists $\epsilon > 0$ such that

$$\frac{1}{2\eta_t} \frac{d}{dt} |\theta_t|^2 \le -\Gamma |\theta_t|^2 + \frac{\epsilon}{2} |\theta_t|^2 + \frac{(|c|_{B_b(S \times A)} + \tau \gamma \mathrm{K}_t)^2}{2\epsilon}$$

$$\le -\Gamma |\theta_t|^2 + \frac{\epsilon}{2} |\theta_t|^2 + \frac{|c|_{B_b(S \times A)}^2 + \tau^2 \gamma^2 \mathrm{K}_t^2}{\epsilon},$$

where we used the identity $(a + b)^2 \le 2a^2 + 2b^2$. Choosing $\epsilon = \Gamma$ we arrive at

$$\frac{1}{2\eta_t} \frac{d}{dt} |\theta_t|^2 \le -\frac{\Gamma}{2} |\theta_t|^2 + \frac{\tau^2 \gamma^2 \mathrm{K}_t^2}{\Gamma} + \frac{|c|_{B_b(S \times A)}^2}{\Gamma}$$

which concludes the proof. $\qquad\square$

## C.2 PROOF OF THEOREM 5.1

*Proof.* By Lemma 5.1, we have that for all $r \geq 0$

$$\frac{1}{2\eta_r}\frac{d}{dr}|\theta_r|^2 \leq -\frac{\Gamma}{2}|\theta_r|^2 + \frac{\tau^2\gamma^2 \mathrm{K}_r^2}{\Gamma} + \frac{|c|^2_{B_b(S \times A)}}{\Gamma}.$$

Rearranging, it holds that for all $t \geq 0$

$$|\theta_r|^2 \leq -\frac{1}{\Gamma\eta_r}\frac{d}{dr}|\theta_r|^2 + \frac{2|c|^2_{B_b(S \times A)} + 2\tau^2\gamma^2 \mathrm{K}_r^2}{\Gamma^2}.$$

Multiplying both sides by $e^{-\tau(t-r)}$ and integrating over $r$ from 0 to $t$ we have that for all $t \geq 0$

$$\int_0^t e^{-\tau(t-r)}|\theta_r|^2 dr \leq -\frac{1}{\Gamma}\int_0^t e^{-\tau(t-r)}\frac{1}{\eta_r}\frac{d}{dr}|\theta_r|^2 dr + \frac{2|c|^2_{B_b(S \times A)}}{\Gamma^2}\int_0^t e^{-\tau(t-r)}dr \qquad (18)$$

$$+ \frac{2\tau^2\gamma^2}{\Gamma^2}\int_0^t e^{-\tau(t-r)}\mathrm{K}_r^2 dr$$

$$\leq -\frac{1}{\Gamma}\int_0^t e^{-\tau(t-r)}\frac{1}{\eta_r}\frac{d}{dr}|\theta_r|^2 dr + \frac{2|c|^2_{B_b(S \times A)}}{\Gamma^2\tau} + \frac{2\tau^2\gamma^2}{\Gamma^2}\int_0^t e^{-\tau(t-r)}\mathrm{K}_r^2 dr,$$

where we used that $\int_0^t e^{-\tau(t-r)}dr \leq \frac{1}{\tau}$. Integrating the first term by parts, we have

$$-\int_0^t e^{-\tau(t-r)}\frac{1}{\eta_r}\frac{d}{dr}|\theta_r|^2 dr = -\frac{|\theta_t|^2}{\eta_t} + e^{-\tau t}\frac{|\theta_0|^2}{\eta_0} + \tau\int_0^t |\theta_r|^2 \frac{e^{-\tau(t-r)}}{\eta_r}dr \qquad (19)$$

$$-\int_0^t |\theta_r|^2 \frac{e^{-\tau(t-r)}\frac{d}{dr}\eta_r}{\eta_r^2}dr.$$

Since by definition we have that for all $t \geq 0$, $\eta_t \geq 1$ and $\frac{d}{dt}\eta_t \geq 0$ it holds that

$$\int_0^t |\theta_r|^2 \frac{e^{-\tau(t-r)}\frac{d}{dr}\eta_r}{\eta_r^2}dr \geq 0.$$

Hence dropping the negative terms on the right hand side of (19) and using that $\eta_t \geq \eta_0$ for all $t \geq 0$, we have

$$-\frac{1}{\Gamma}\int_0^t e^{-\tau(t-r)}\frac{1}{\eta_r}\frac{d}{dr}|\theta_r|^2 dr \leq e^{-\tau t}\frac{|\theta_0|^2}{\Gamma\eta_0} + \frac{\tau}{\Gamma\eta_0}\int_0^t e^{-\tau(t-r)}|\theta_r|^2 dr.$$

Substituting this back into (18), for all $t \geq 0$ we have that

$$\int_0^t e^{-\tau(t-r)}|\theta_r|^2 dr \leq e^{-\tau t}\frac{|\theta_0|^2}{\Gamma\eta_0} + \frac{\tau}{\Gamma\eta_0}\int_0^t e^{-\tau(t-r)}|\theta_r|^2 dr$$

$$+ \frac{2|c|^2_{B_b(S \times A)}}{\Gamma^2\tau} + \frac{2\tau^2\gamma^2}{\Gamma^2}\int_0^t e^{-\tau(t-r)}\mathrm{K}_r^2 dr.$$

Grouping like terms we have

$$\left(1 - \frac{\tau}{\Gamma\eta_0}\right)\int_0^t e^{-\tau(t-r)}|\theta_r|^2 dr \leq e^{-\tau t}\frac{|\theta_0|^2}{\Gamma\eta_0} + \frac{2|c|^2_{B_b(S \times A)}}{\Gamma^2\tau} + \frac{2\tau^2\gamma^2}{\Gamma^2}\int_0^t e^{-\tau(t-r)}\mathrm{K}_r^2 dr.$$

Recall that we have $\eta_0 > \frac{\tau}{\Gamma}$ to ensure that $1 - \frac{\tau}{\Gamma\eta_0} > 0$. Dividing through by $1 - \frac{\tau}{\Gamma\eta_0}$ gives for all $t \geq 0$ that

$$\int_0^t e^{-\tau(t-r)}|\theta_r|^2 dr \leq \sigma_1 + \sigma_2 \int_0^t e^{-\tau(t-r)}\mathrm{K}_r^2 dr \qquad (20)$$

where we've set

$$\sigma_1 := \frac{|\theta_0|^2}{\Gamma\eta_0\left(1 - \frac{\tau}{\Gamma\eta_0}\right)} + \frac{2|c|^2_{B_b(S \times A)}}{\Gamma^2\tau\left(1 - \frac{\tau}{\Gamma\eta_0}\right)},$$

$$\sigma_2 := \frac{2\tau^2\gamma^2}{\Gamma^2\left(1 - \frac{\tau}{\Gamma\eta_0}\right)}.$$

Recall the approximate Fisher–Rao gradient flow for the policies $\{\pi_t\}_{t\geq 0}$, which for all $t \geq 0$ and for all $s \in S$, $a \in A$ is

$$\partial_t \ln\frac{d\pi_t}{d\mu}(s,a) = -\left(Q_t(s,a) + \tau\ln\frac{d\pi_t}{d\mu}(a,s) - \int_A\left(Q_t(s,a) + \tau\ln\frac{d\pi_t}{d\mu}(a,s)\right)\pi_t(da|s)\right)$$

Duhamel's principle yields for all $t \geq 0$ that

$$\ln\frac{d\pi_t}{d\mu}(s,a) = e^{-\tau t}\ln\frac{d\pi_0}{d\mu}(a,s) + \int_0^t e^{-\tau(t-r)}\left(\int_A Q_r(s,a)\pi_r(da|s) - Q_r(s,a)\right)dr \quad (21)$$

$$+ \tau\int_0^t e^{-\tau(t-r)}\,\mathrm{KL}(\pi_r(\cdot|s)|\mu)dr$$

Observe that since $\pi_0 \in \Pi_\mu$, there exists $C_1 \geq 1$ such that $\ln\left|\frac{d\pi_t}{d\mu}\right|_{B_b(S\times A)} \leq C_1$. Using that $e^{-\tau t} \leq 1$ and assumption 4.2 gives that for all $t \geq 0$

$$\ln\frac{d\pi_t}{d\mu}(s,a) \leq C_1 + 2\int_0^t e^{-\tau(t-r)}|\theta_r|dr + \tau\int_0^t e^{-\tau(t-r)}\,\mathrm{KL}(\pi_r(\cdot|s)|\mu)dr$$

$$\leq C_1 + 2\int_0^t e^{-\tau(t-r)}|\theta_r|dr + \tau\int_0^t e^{-\tau(t-r)}\mathrm{K}_r dr$$

Integrating over the actions with respect to $\pi_t(\cdot|s) \in \mathcal{P}(A)$ gives for all $t \geq 0$ that

$$\mathrm{KL}(\pi_t(\cdot|s)|\mu) \leq C_1 + 2\int_0^t e^{-\tau(t-r)}|\theta_r|dr + \tau\int_0^t e^{-\tau(t-r)}\mathrm{K}_r dr$$

where we again use that $\mathrm{K}_r = \sup_{s\in S}\mathrm{KL}(\pi_r(\cdot|s)|\mu)$. Following from the techniques in Liu et al. (2023), observe that from (21) and Assumption 4.2 we similarly get for all $t \geq 0$ that

$$\ln\frac{d\mu}{d\pi_t}(a,s) = -\ln\frac{d\pi_t}{d\mu}(s,a) \leq C_1 + 2\int_0^t e^{-\tau(t-r)}|\theta_r|dr - \tau\int_0^t e^{-\tau(t-r)}\mathrm{K}_r dr.$$

Now integrating over the actions with respect to the reference measure $\mu \in \mathcal{P}(A)$ we have

$$\mathrm{KL}(\mu|\pi_t(\cdot|s)) \leq C_1 + 2\int_0^t e^{-\tau(t-r)}|\theta_r|dr - \tau\int_0^t e^{-\tau(t-r)}\mathrm{K}_r dr$$

Moreover, using the non-negativity of the KL divergence, it holds for all $t \geq 0$ that

$$\mathrm{KL}(\pi_t(\cdot|s)|\mu) \leq \mathrm{KL}(\pi_t(\cdot|s)|\mu) + \mathrm{KL}(\mu|\pi_t(\cdot|s)) \leq 2C_1 + 4\int_0^t e^{-\tau(t-r)}|\theta_r|dr$$

Since this holds for any $s \in S$, it holds for all $t \geq 0$ that

$$\mathrm{K}_t \leq 2C_1 + 4\int_0^t e^{-\tau(t-r)}|\theta_r|dr$$

Now squaring both sides and using the Hölder's inequality, we have

$$\mathrm{K}_t^2 \leq \left(2C_1 + 4\int_0^t e^{-\tau(t-r)}|\theta_r|dr\right)^2$$

$$\leq 8(C_1)^2 + 32\left(\int_0^t e^{-\tau(t-r)}|\theta_r|dr\right)^2$$

$$= 8(C_1)^2 + 32\left(\int_0^t e^{-\frac{\tau}{2}(t-r)}e^{-\frac{\tau}{2}(t-r)}|\theta_r|dr\right)^2$$

$$\leq 8(C_1)^2 + 32\left(\int_0^t e^{-\tau(t-r)}dr\right)\left(\int_0^t e^{-\tau(t-r)}|\theta_r|^2 dr\right)$$

$$\leq 8(C_1)^2 + \frac{32}{\tau}\int_0^t e^{-\tau(t-r)}|\theta_r|^2 dr, \quad (22)$$

where we again used $\int_0^t e^{-\tau(t-r)} dr \leq \frac{1}{\tau}$. We can now substitute (20) into (22) to arrive at

$$\mathrm{K}_t^2 \leq 8(C_1)^2 + \frac{32}{\tau}\sigma_1 + \frac{32}{\tau}\sigma_2 \int_0^t e^{-\tau(t-r)} \mathrm{K}_r^2 dr$$

$$:= a_1 + a_2 \int_0^t e^{-\tau(t-r)} \mathrm{K}_r^2 dr$$

with $a_1 = 8(C_1)^2 + \frac{32}{\tau}\sigma_1$ and $a_2 = \frac{32\sigma_2}{\tau}$. $\qquad\square$

### C.3 PROOF OF COROLLARY 5.1

*Proof.* By Theorem 5.1 it holds that

$$\mathrm{K}_t^2 \leq a_1 + a_2 \int_0^t e^{-\tau(t-r)} \mathrm{K}_r^2 dr.$$

Observe that by multiplying through by $e^{\tau t}$, we can rewrite this as

$$e^{\tau t} \mathrm{K}_t^2 \leq e^{\tau t} a_1 + a_2 \int_0^t e^{\tau r} \mathrm{K}_r^2 dr.$$

Hence after defining $g(t) = e^{\tau t} \mathrm{K}_t^2$ and applying Grönwall's inequality (Lemma A.3), for all $\gamma \in (0, 1)$ it holds for all $t \geq 0$ that

$$\mathrm{K}_t^2 \leq a_1 e^{a_2 t}.$$

$\qquad\square$

### C.4 PROOF OF COROLLARY 5.2

*Proof.* By Corollary 5.1 and Lemma 5.1, for all $\gamma \in (0, 1)$ it holds that

$$\frac{1}{2}\frac{d}{dt}|\theta_t|^2 \leq -\frac{\Gamma}{2}\eta_t|\theta_t|^2 + b_t\eta_t$$

such that

$$b_t = \left( \frac{2|c|^2_{B_b(S \times A)} + 2\tau^2\gamma^2 a_1 e^{a_2 t}}{\Gamma^2} \right).$$

Recall that there exists $\alpha > 0$ such that $\frac{d}{dt}\eta_t \leq \alpha\eta_t$, then another application of Grönwall's Lemma then concludes the proof. $\qquad\square$

## D PROOF OF CONVERGENCE RESULTS

### D.1 PROOF OF LEMMA 6.1

*Proof.* By the definition of the state-action value function (2) it holds that

$$\frac{d}{dt}Q_\tau^{\pi_t}(s, a) = \lim_{h \to 0} \frac{Q_\tau^{\pi_{t+h}}(s, a) - Q_\tau^{\pi_t}(s, a)}{h}$$

$$= \gamma \int_S \frac{d}{dt}V_\tau^{\pi_t}(s')P(ds'|s, a).$$

Now observe that by Kerimkulov et al. (2025a)[Proof of Proposition 2.6], we have

$$\frac{d}{dt}V_\tau^{\pi_t}(s) = \frac{1}{1-\gamma}\int_{S \times A} A_\tau^{\pi_t}(s, a)\partial_t\pi_t(da|s')d^{\pi_t}(ds'|s).$$

Thus we have

$$\frac{d}{dt}Q_\tau^{\pi_t}(s, a) = \frac{\gamma}{1-\gamma}\int_S \left( \int_{S \times A} A_\tau^{\pi_t}(s'', a'')\partial_t\pi_t(da''|s'')d^{\pi_t}(ds''|s') \right) P(ds'|s, a).$$

$\qquad\square$

### D.2 Proof of Theorem 6.1

*Proof.* Recall the performance difference Lemma (Lemma A.2): for all $\rho \in \mathcal{P}(S)$ and $\pi, \pi' \in \Pi_\mu$,

$$V_\tau^\pi(\rho) - V_\tau^{\pi'}(\rho)$$
$$= \frac{1}{1-\gamma} \int_S \left[ \int_A \left( Q_\tau^{\pi'}(s,a) + \tau \ln \frac{d\pi'}{d\mu}(a,s) \right)(\pi - \pi')(da|s) + \tau \, \mathrm{KL}(\pi(\cdot|s)|\pi'(\cdot|s)) \right] d_\rho^\pi(ds).$$

Now let $\pi = \pi^*$ and $\pi' = \pi_t$ and multiply both sides by $-1$ we have

$$V_\tau^{\pi_t}(\rho) - V_\tau^{\pi^*}(\rho) = \frac{-1}{1-\gamma} \int_S \left( \int_A \left( Q^{\pi_t}(s,a) + \tau \ln \frac{d\pi_t}{d\mu}(a,s) \right)(\pi^* - \pi_t)(da|s) \right. \tag{23}$$
$$\left. + \tau \, \mathrm{KL}(\pi^*(\cdot|s)|\pi_t(\cdot|s)) \right) d_\rho^{\pi^*}(ds).$$

Recall the approximate Fisher–Rao dynamics, which we write as

$$\partial_t \ln \frac{d\pi_t}{d\mu}(s,a) + \left( Q_t(s,a) + \tau \ln \frac{d\pi_t}{d\mu}(a,s) - \int_A \left( Q_t(s,a') + \tau \ln \frac{d\pi_t}{d\mu}(a',s) \right) \pi_t(da'|s) \right) = 0. \tag{24}$$

Observe that since the normalisation constant (enforcing the conservation of mass along the flow) $\int_A \left( Q_t(s,a) + \tau \ln \frac{d\pi_t}{d\mu}(a,s) \right) \pi_t(da|s)$ is independent of $a \in A$, it holds that

$$\int_A \left( \int_A \left( Q_t(s,a') + \tau \ln \frac{d\pi_t}{d\mu}(a',s) \right) \pi_t(da'|s) \right)(\pi^* - \pi_t)(da|s) = 0.$$

Hence adding 0 in the form of (24) into (23) it holds that for all $t \geq 0$

$$V_\tau^{\pi_t}(\rho) - V_\tau^{\pi^*}(\rho) = \frac{1}{1-\gamma} \left( \int_{S \times A} \partial_t \ln \frac{d\pi_t}{d\mu}(a,s)(\pi^* - \pi_t)(da|s) d_\rho^{\pi^*}(ds) \right. \tag{25}$$
$$\left. + \int_{S \times A} (Q_t(s,a) - Q^{\pi_t}(s,a))(\pi^* - \pi_t)(da|s) d_\rho^{\pi^*}(ds) - \tau \int_S \mathrm{KL}(\pi^*(\cdot|s)|\pi_t(\cdot|s) d_\rho^{\pi^*}(ds) \right).$$

By (Kerimkulov et al., 2025b, Lemma 3.8) and Corollary 5.1, for any fixed $\nu \in \Pi_\mu$, the map $t \to \mathrm{KL}(\nu|\pi_t)$ is differentiable. Hence we have

$$\int_A \partial_t \ln \frac{d\pi_t}{d\mu}(s,a)(\pi^* - \pi_t)(da|s) = \int_A \partial_t \ln \frac{d\pi_t}{d\mu}(s,a)\pi^*(da|s) - \int_A \partial_t \ln \frac{d\pi_t}{d\mu}(s,a)\pi_t(da|s)$$
$$= \int_A \partial_t \ln \frac{d\pi_t}{d\mu}(s,a)\pi^*(da|s)$$
$$= -\frac{d}{dt} \mathrm{KL}(\pi^*(\cdot|s)|\pi_t(\cdot|s)),$$

where we used the conservation of mass of the policy dynamics in the second equality. Substituting this into (25) we have

$$V_\tau^{\pi_t}(\rho) - V_\tau^{\pi^*}(\rho) = \frac{1}{1-\gamma} \left( -\frac{d}{dt} \int_S \mathrm{KL}(\pi^*(\cdot|s)|\pi_t(\cdot|s)) d_\rho^{\pi^*}(ds) \right. \tag{26}$$
$$\left. + \int_{S \times A} (Q_t(s,a) - Q^{\pi_t}(s,a))(\pi^* - \pi_t)(da|s) d_\rho^{\pi^*}(ds) - \tau \int_S \mathrm{KL}(\pi^*(\cdot|s)|\pi_t(\cdot|s) d_\rho^{\pi^*}(ds) \right).$$

Focusing on the second term, we have

$$\int_{S \times A} (Q_t(s,a) - Q^{\pi_t}(s,a))(\pi^* - \pi_t)(da|s)d_\rho^{\pi^*}(ds)$$

$$\leq |Q_t(s,a) - Q^{\pi_t}(s,a)|_{B_b(S \times A)} \int_S \mathrm{TV}(\pi^*(\cdot|s), \pi_t(\cdot|s))d_\rho^{\pi^*}(ds)$$

$$\leq \frac{1}{\sqrt{2}}|\theta_t - \theta_{\pi_t}| \int_S \mathrm{KL}(\pi^*(\cdot|s)|\pi_t(\cdot|s))^{\frac{1}{2}}d_\rho^{\pi^*}(ds)$$

$$\leq \frac{1}{\sqrt{2}}|\theta_t - \theta_{\pi_t}| \left( \int_S \mathrm{KL}(\pi^*(\cdot|s)|\pi_t(\cdot|s))d_\rho^{\pi^*}(ds) \right)^{\frac{1}{2}},$$

where we used Pinsker's Inequality in the second inequality and Hölder's inequality in the final inequality. Now applying Young's inequality, there exists $\epsilon > 0$ such that

$$|\theta_t - \theta_{\pi_t}| \left( \int_S \mathrm{KL}(\pi^*(\cdot|s)|\pi_t(\cdot|s))d_\rho^{\pi^*}(ds) \right)^{\frac{1}{2}} \leq \frac{1}{2\epsilon}|\theta_t - \theta_{\pi_t}|^2 + \frac{\epsilon}{2} \int_S \mathrm{KL}(\pi^*(\cdot|s)|\pi_t(\cdot|s))d_\rho^{\pi^*}(ds).$$

Substituting this back into (26) and choosing $\epsilon = \sqrt{2}\tau$ we have

$$V_\tau^{\pi_t}(\rho) - V_\tau^{\pi^*}(\rho) = \frac{1}{1-\gamma}\left( -\frac{d}{dt}\int_S \mathrm{KL}(\pi^*(\cdot|s)|\pi_t(\cdot|s))d_\rho^{\pi^*}(ds) \right.$$

$$\left. -\frac{\tau}{2}\int_S \mathrm{KL}(\pi^*(\cdot|s)|\pi_t(\cdot|s)d_\rho^{\pi^*}(ds) + \frac{1}{4\tau}|\theta_t - \theta_{\pi_t}|^2 \right).$$

Rearranging, we arrive at

$$\frac{d}{dt}\int_S \mathrm{KL}(\pi^*(\cdot|s)|\pi_t(\cdot|s))d_\rho^{\pi^*}(ds) \leq -\frac{\tau}{2}\int_S \mathrm{KL}(\pi^*(\cdot|s)|\pi_t(\cdot|s))d_\rho^{\pi^*}(ds)$$

$$- (1-\gamma)\left(V_\tau^{\pi_t}(\rho) - V_\tau^{\pi^*}(\rho)\right) + \frac{1}{4\tau}|\theta_t - \theta_{\pi_t}|^2.$$

Applying Duhamel's principle yields

$$\int_S \mathrm{KL}(\pi^*(\cdot|s)|\pi_t(\cdot|s))d_\rho^{\pi^*}(ds) \leq e^{-\frac{\tau}{2}t}\int_S \mathrm{KL}(\pi^*(\cdot|s)|\pi_0(\cdot|s))d_\rho^{\pi^*}(ds)$$

$$- (1-\gamma)\int_0^t e^{-\frac{\tau}{2}(t-r)}(V_\tau^{\pi_r}(\rho) - V_\tau^{\pi^*}(\rho))dr + \frac{1}{2\tau}\int_0^t e^{-\frac{\tau}{2}(t-r)}|\theta_r - \theta_{\pi_r}|^2 dr.$$

Now using that $\int_0^t e^{-\frac{\tau}{2}(t-r)}dr = \frac{2(1-e^{-\frac{\tau}{2}})}{\tau}$, we have

$$\int_S \mathrm{KL}(\pi^*(\cdot|s)|\pi_t(\cdot|s))d_\rho^{\pi^*}(ds) \leq e^{-\frac{\tau}{2}t}\int_S \mathrm{KL}(\pi^*(\cdot|s)|\pi_0(\cdot|s))d_\rho^{\pi^*}(ds)$$

$$- \frac{2(1-\gamma)(1-e^{-\frac{\tau}{2}})}{\tau}\min_{r \in [0,t]}\left(V_\tau^{\pi_r}(\rho) - V_\tau^{\pi^*}(\rho)\right) + \frac{1}{2\tau}\int_0^t e^{-\frac{\tau}{2}(t-r)}|\theta_r - \theta_{\pi_r}|^2 dr.$$

Rearranging, we have

$$\min_{r \in [0,t]} V_\tau^{\pi_r}(\rho) - V_\tau^{\pi^*}(\rho) \leq \frac{\tau}{2(1-\gamma)(1-e^{-\frac{\tau}{2}})}\left( e^{-\frac{\tau}{2}t}\int_S \mathrm{KL}(\pi^*(\cdot|s)|\pi_0(\cdot|s))d_\rho^{\pi^*}(ds) \right.$$

$$\left. + \frac{1}{2\tau}\int_0^t e^{-\frac{\tau}{2}(t-r)}|\theta_r - \theta_{\pi_r}|^2 dr \right).$$

which concludes the proof.

$\square$

### D.3 PROOF OF THEOREM 6.2

*Proof.* Using the chain rule and the critic dynamics in (8), we have that for all $r \geq 0$

$$\frac{1}{2\eta_r}\frac{d}{dr}|\theta_r - \theta_{\pi_r}|^2 = \frac{1}{\eta_r}\left(\left\langle \frac{d\theta_r}{dr}, \theta_r - \theta_{\pi_r}\right\rangle - \left\langle \frac{d\theta_{\pi_r}}{dr}, \theta_r - \theta_{\pi_r}\right\rangle\right)$$

$$= -\langle g(\theta_r, \pi_r), \theta_r - \theta_{\pi_r}\rangle - \frac{1}{\eta_r}\left\langle \frac{d\theta_{\pi_r}}{dr}, \theta_r - \theta_{\pi_r}\right\rangle$$

Let $\Gamma = \lambda_\beta(1-\gamma)(1-\sqrt{\gamma})$. Using Lemma 4.1 and the $\lambda_\beta$-strong convexity of $L(\cdot, \pi; \beta)$ and recalling that $L(\theta_{\pi_r}, \pi_r) = 0$ for all $r \geq 0$, it holds for all $r \geq 0$ that

$$\frac{1}{2\eta_t}\frac{d}{dt}|\theta_t - \theta_{\pi_t}|^2 = -\langle g(\theta_t, \pi_t), \theta_t - \theta_{\pi_t}\rangle - \frac{1}{\eta_t}\left\langle \frac{d\theta_{\pi_t}}{dt}, \theta_t - \theta_{\pi_t}\right\rangle$$

$$\leq -(1-\gamma)(1-\sqrt{\gamma})\langle \nabla_\theta L(\theta_t, \pi_t; \beta), \theta_t - \theta_{\pi_t}\rangle - \frac{1}{\eta_t}\left\langle \frac{d\theta_{\pi_t}}{dt}, \theta_t - \theta_{\pi_t}\right\rangle$$

$$\leq -(1-\gamma)(1-\sqrt{\gamma})L(\theta_t, \pi_t; \beta) - \frac{\Gamma}{2}|\theta_t - \theta_{\pi_t}|^2 - \frac{1}{\eta_t}\left\langle \frac{d\theta_{\pi_t}}{dt}, \theta_t - \theta_{\pi_t}\right\rangle$$

$$\leq -(1-\gamma)(1-\sqrt{\gamma})L(\theta_t, \pi_t; \beta) - \frac{\Gamma}{2}|\theta_t - \theta_{\pi_t}|^2 + \frac{1}{2\eta_t}\left(\left|\frac{d\theta_{\pi_t}}{dt}\right|^2 + |\theta_t - \theta_{\pi_t}|^2\right)$$

$$\tag{27}$$

$$= -(1-\gamma)(1-\sqrt{\gamma})L(\theta_t, \pi_t; \beta) - \left(\frac{\Gamma}{2} - \frac{1}{2\eta_t}\right)|\theta_t - \theta_{\pi_t}|^2 + \frac{1}{2\eta_t}\left|\frac{d\theta_{\pi_t}}{dt}\right|^2,$$

where we used Hölder's and Young's inequalities in (27). Since $\eta_0 > \frac{1}{\Gamma}$ and $\eta_t$ is a non-decreasing function, it holds that $\eta_t > \frac{1}{\Gamma}$ for all $t \geq 0$. Hence $\frac{\Gamma}{2} - \frac{1}{2\eta_t} > 0$ and thus we can drop the second term. Moreover the $\lambda_\beta$-strong convexity of $L(\cdot, \pi; \beta)$ along with $L(\theta_\pi, \pi; \beta) = 0$ and $\nabla_\theta L(\theta_\pi, \pi) = 0$ for all $\pi \in \Pi_\mu$ gives that

$$|\theta_t - \theta_{\pi_t}|^2 \leq \frac{2}{\lambda_\beta}L(\theta_t, \pi_t; \beta).$$

Hence for all $r \geq 0$ we arrive at

$$\frac{1}{2\eta_r}\frac{d}{dr}|\theta_r - \theta_{\pi_r}|^2 \leq -\frac{\Gamma}{2}|\theta_r - \theta_{\pi_r}|^2 + \frac{1}{2\eta_r}\left|\frac{d\theta_{\pi_r}}{dr}\right|^2.$$

Rearranging, multiplying by $e^{-\tau(t-r)}$ and integrating over $r$ from 0 to $t$, it holds for all $t \geq 0$ that

$$\int_0^t e^{-\frac{\tau}{2}(t-r)}|\theta_r - \theta_{\pi_r}|^2 dr \leq -\frac{1}{\Gamma}\int_0^t e^{-\frac{\tau}{2}(t-r)}\frac{1}{\eta_r}\frac{d}{dr}|\theta_r - \theta_{\pi_r}|^2 dr + \frac{1}{\Gamma}\int_0^t e^{-\frac{\tau}{2}(t-r)}\frac{1}{\eta_r}\left|\frac{d\theta_{\pi_r}}{dt}\right|^2 dr.$$

Integrating the first term by parts (identically to (19) from the proof of Theorem 5.1), we have

$$\int_0^t e^{-\frac{\tau}{2}(t-r)}|\theta_r - \theta_{\pi_r}|^2 dr \leq \frac{1}{\Gamma}\left(-\frac{|\theta_t - \theta_{\pi_t}|^2}{\eta_t} + e^{-\frac{\tau}{2}t}\frac{|\theta_0 - \theta_{\pi_0}|^2}{\eta_0}\right.$$

$$+ \frac{\tau}{2}\int_0^t e^{-\frac{\tau}{2}(t-r)}\frac{1}{\eta_r}|\theta_r - \theta_{\pi_r}|^2 dr - \int_0^t |\theta_r - \theta_{\pi_r}|^2\frac{e^{-\frac{\tau}{2}(t-r)}\frac{d}{dr}\eta_r}{\eta_r^2} dr$$

$$\left. + \int_0^t e^{-\frac{\tau}{2}(t-r)}\frac{1}{\eta_r}\left|\frac{d\theta_{\pi_r}}{dr}\right|^2 dr\right).$$

Since for all $t \geq 0$ it holds that $\eta_t \geq 1$ and $\frac{d}{dt}\eta_t \geq 0$, we have that

$$\int_0^t |\theta_r - \theta_{\pi_r}|^2\frac{e^{-\frac{\tau}{2}(t-r)}\frac{d}{dr}\eta_r}{\eta_r^2} dr \geq 0.$$

Thus after dropping all negative terms and using that $\eta_t \geq \eta_0$ for all $t \geq 0$, we have

$$\left(1 - \frac{\tau}{2\Gamma\eta_0}\right) \int_0^t e^{-\frac{\tau}{2}(t-r)} |\theta_r - \theta_{\pi_r}|^2 dr \leq e^{-\frac{\tau}{2}} \frac{|\theta_0 - \theta_{\pi_0}|^2}{\Gamma\eta_0} + \int_0^t e^{-\frac{\tau}{2}(t-r)} \frac{1}{\eta_r} \left|\frac{d\theta_{\pi_r}}{dr}\right|^2 dr.$$

Since $\eta_0 > \frac{1}{2\Gamma}$ and $\tau < 1$, it holds that $1 - \frac{\tau}{2\Gamma\eta_0} > 0$ and hence it holds that

$$\int_0^t e^{-\frac{\tau}{2}(t-r)} |\theta_r - \theta_{\pi_r}|^2 dr \leq e^{-\frac{\tau}{2}} \frac{|\theta_0 - \theta_{\pi_0}|^2}{\Gamma\eta_0 \left(1 - \frac{\tau}{2\Gamma\eta_0}\right)} + \frac{1}{\left(1 - \frac{\tau}{2\Gamma\eta_0}\right)} \int_0^t e^{-\frac{\tau}{2}(t-r)} \frac{1}{\eta_r} \left|\frac{d\theta_{\pi_r}}{dr}\right|^2 dr,$$

which concludes the proof. □

### D.4 Proof of Theorem 6.3

*Proof.* By Theorem 6.2, we have

$$\int_0^t e^{-\frac{\tau}{2}(t-r)} |\theta_r - \theta_{\pi_r}|^2 dr \leq e^{-\frac{\tau}{2}} \frac{|\theta_0 - \theta_{\pi_0}|^2}{\Gamma\eta_0 \left(1 - \frac{\tau}{2\Gamma\eta_0}\right)} + \frac{1}{\left(1 - \frac{\tau}{2\Gamma\eta_0}\right)} \int_0^t e^{-\frac{\tau}{2}(t-r)} \frac{1}{\eta_r} \left|\frac{d\theta_{\pi_r}}{dr}\right|^2 dr.$$

Hence it remains to characterise the growth of the final integral. Observe that for all $\pi \in \mathcal{P}(A|S)$, $\theta_\pi \in \mathbb{R}^N$ satisfies the least-squares optimality condition given by

$$\theta_\pi = \arg\min_\theta L(\theta, \pi; \beta) = \left(\int_{S \times A} \phi(s,a)\phi(s,a)^\top \beta(da,ds)\right)^{-1} \left(\int_{S \times A} \phi(s,a) Q_\tau^\pi(s,a)\, \beta(ds,da)\right).$$

Setting $\pi = \pi_t$ and differentiating time we arrive at

$$\frac{d\theta_{\pi_t}}{dt} = \left(\int_{S \times A} \phi(s,a)\phi(s,a)^\top \beta(da,ds)\right)^{-1} \left(\int_{S \times A} \phi(s,a) \frac{d}{dt} Q^{\pi_t}(s,a)\, \beta(ds,da)\right).$$

Hence by Lemma 6.1, Assumption 4.2 and Assumption 4.3, for all $t \geq 0$ it holds that

$$\left|\frac{d\theta_{\pi_t}}{dt}\right| = \left|\left(\int_{S \times A} \phi(s,a)\phi(s,a)^\top \beta(da,ds)\right)^{-1} \left(\int_{S \times A} \phi(s,a) \frac{d}{dt} Q_\tau^{\pi_t}(s,a)\, \beta(ds,da)\right)\right|$$

$$\leq \left|\left(\int_{S \times A} \phi(s,a)\phi(s,a)^\top \beta(da,ds)\right)^{-1}\right|_{\mathrm{op}} \left|\frac{d}{dt} Q_\tau^{\pi_t}\right|_{B_b(S \times A)}$$

$$= \frac{1}{\lambda_\beta} \left|\frac{d}{dt} Q^{\pi_t}\right|_{B_b(S \times A)}$$

$$= \frac{\gamma}{\lambda_\beta(1-\gamma)} \left|\int_S \left(\int_{S \times A} A_\tau^{\pi_t}(s'',a'') \partial_t \pi_t(da''|s'')\, d^{\pi_t}(ds''|s')\right) P(ds'|\cdot,\cdot)\right|_{B_b(S \times A)}$$

$$\leq \frac{\gamma}{\lambda_\beta(1-\gamma)} |A_\tau^{\pi_t}|_{B_b(S \times A)} \sup_{s \in S} |\partial_t \pi_t(\cdot|s)|_{\mathcal{M}(A)}.$$

Now using Lemma B.2, it holds that

$$|A_\tau^{\pi_t}|_{B_b(S \times A)} \sup_{s \in S} |\partial_t \pi_t(\cdot|s)|_{\mathcal{M}(A)} \leq |A_\tau^{\pi_t}|_{B_b(S \times A)} |A_t|_{B_b(S \times A)}$$

$$\leq \left(2|Q_\tau^{\pi_t}|_{B_b(S \times A)} + 2\tau \left|\ln \frac{d\pi_t}{d\mu}\right|_{B_b(S \times A)}\right) \left(2|Q_t|_{B_b(S \times A)} + 2\tau \left|\ln \frac{d\pi_t}{d\mu}\right|_{B_b(S \times A)}\right).$$

Hence by Corollaries 5.1 and 5.2 and Lemma B.2, there exists $\alpha_1, \alpha_2 > 0$ such that

$$\left|\frac{d\theta_{\pi_t}}{dt}\right|^2 \leq \alpha_1 e^{\alpha_2 t}.$$

Thus Theorem 6.2 becomes

$$\int_0^t e^{-\frac{\tau}{2}(t-r)} |\theta_r - \theta_{\pi_r}|^2 dr \leq e^{-\frac{\tau}{2}} \frac{|\theta_0 - \theta_{\pi_0}|^2}{\Gamma\eta_0 \left(1 - \frac{\tau}{2\Gamma\eta_0}\right)} + \frac{\alpha_1}{\left(1 - \frac{\tau}{2\Gamma\eta_0}\right)} \int_0^t e^{-\frac{\tau}{2}(t-r)} \frac{e^{\alpha_2 r}}{\eta_r} dr.$$

Let $\eta_t = \eta_0 e^{k_1 t}$ for any $k_1 > \frac{\tau}{2} + \alpha_2$. Then observe that

$$\int_0^t e^{-\frac{\tau}{2}(t-r)} \frac{e^{\alpha_2 r}}{\eta_r} dr = \frac{1}{\eta_0} e^{-\frac{\tau}{2}t} \int_0^t e^{\left(\frac{\tau}{2} + \alpha_2 - k_1\right)r} dr$$

$$\leq \frac{1}{\eta_0} e^{-\frac{\tau}{2}t} \left( \frac{e^{\left(\frac{\tau}{2} + \alpha_2 - k_1\right)t} - 1}{\frac{\tau}{2} + \alpha_2 - k_1} \right)$$

$$\leq \frac{e^{-\frac{\tau}{2}t}}{\eta_0 \left( \frac{\tau}{2} + \alpha_2 - k_1 \right)},$$

hence all together it holds that

$$\int_0^t e^{-\frac{\tau}{2}(t-r)} |\theta_r - \theta_{\pi_r}|^2 dr \leq e^{-\frac{\tau}{2}} \frac{|\theta_0 - \theta_{\pi_0}|^2}{\Gamma \eta_0 \left( 1 - \frac{\tau}{2\Gamma \eta_0} \right)} + e^{-\frac{\tau}{2}t} \frac{\alpha_1}{\left( \eta_0 - \frac{\tau}{2\Gamma} \right) \left( \frac{\tau}{2} + \alpha_2 - k_1 \right)}.$$

Substituting this into the result from Theorem 6.2 concludes the proof. □

## E    ADDITIONAL RESULTS

**Corollary E.1** (Uniform boundedness). *Under the same assumptions as Theorem 5.1, for $\gamma \in (0,1)$ such that $\frac{64\gamma^2}{\Gamma^2 - \frac{\Gamma\tau}{\eta_0}} < 1$ it holds that $a_2 < \tau$ and for all $t \geq 0$ it holds that*

$$\mathrm{KL}(\pi_t(\cdot|s)\|\mu)^2 \leq \frac{a_1 \tau}{\tau - a_2}$$

### E.1    PROOF OF COROLLARY E.1

*Proof.* By Theorem 5.1 we have that

$$\mathrm{K}_t^2 \leq a_1 + a_2 \int_0^t e^{-\tau(t-r)} \mathrm{K}_r^2 dr.$$

Taking the supremum over $[0, t]$ on the right hand side, we have

$$\mathrm{K}_t^2 \leq a_1 + \frac{a_2}{\tau} \sup_{r \in [0,t]} \mathrm{K}_r^2.$$

Since this holds for all $t \geq 0$, we have

$$\sup_{r \in [0,t]} \mathrm{K}_r^2 \leq a_1 + \frac{a_2}{\tau} \sup_{r \in [0,t]} \mathrm{K}_r^2.$$

Now forcing $1 - \frac{a_2}{\tau} > 0$, which is equivalent to the condition

$$\frac{64\gamma^2}{\Gamma^2 - \frac{\Gamma\tau}{\eta_0}} < 1.$$

Hence after rearranging we have

$$\mathrm{K}_t^2 \leq \sup_{r \in [0,t]} \mathrm{K}_r^2 \leq \frac{a_1 \tau}{\tau - a_2}$$

□

**Remark E.1.** *Observe that if one does not apply the loose upper bound $e^{-\tau t} \leq 1$ in (21) from the proof of Theorem 5.1, it holds that*

$$a_1 = a_1(t) = 8e^{-2\tau t}(C_1)^2 + \frac{32}{\tau}\sigma_1$$

*with $\sigma_1 := \frac{|\theta_0|^2}{\Gamma \eta_0 \left( 1 - \frac{\tau}{\Gamma\eta_0} \right)} + \frac{2|c|^2_{B_b(S \times A)}}{\Gamma^2 \tau \left( 1 - \frac{\tau}{\Gamma\eta_0} \right)}$. Then choosing $\eta_0 = \tau + \epsilon$ for any $\epsilon > 0$ so that the conditions of Theorem 5.1 holds, formally sending $\tau \to \infty$ we obtain $\mathrm{KL}(\pi_t(\cdot|s)\|\mu) \to 0$ for all $s \in S$.*

**Corollary E.2.** *Under the conditions of Corollary E.1, there exists $R > 0$ such that for all $t \geq 0$ it holds that*

$$|\theta_t| \leq R$$

### E.2 PROOF OF COROLLARY E.2

*Proof.* By Corollary E.1, for sufficiently small $\gamma > 0$ it holds that for all $t \geq 0$,

$$\mathrm{K}_t^2 \leq \frac{a_1 \tau}{\tau - a_2}.$$

Hence by Lemma 5.1 we have

$$\frac{1}{2}\frac{d}{dt}|\theta_t|^2 \leq -\eta_t \frac{\Gamma}{2}|\theta_t|^2 + \eta_t \left( \frac{2|c|_{B_b(S \times A)}^2 + 2\tau^2 \gamma^2 \left( \frac{a_1 \tau}{\tau - a_2} \right)}{\Gamma^2} \right).$$

The uniform boundedness in time of $|\theta_t|$ then follows by Grönwall's Lemma (Lemma A.3). $\square$

**Corollary E.3.** *Under the same assumptions as Theorem 6.2, for $\gamma \in (0,1)$ such that $\frac{2\sqrt{2}\gamma}{\sqrt{\Gamma^2 - \frac{\Gamma \tau}{\eta_0}}} < 1$ there exists $d_1 > 0$ such that for all $t \geq 0$,*

$$\min_{r \in [0,t]} V_\tau^{\pi_r}(\rho) - V_\tau^{\pi^*}(\rho) \leq \frac{\tau}{2(1-\gamma)(1 - e^{-\frac{\tau}{2}t})} \left( e^{-\frac{\tau}{2}t} \int_S \mathrm{KL}(\pi^*(\cdot|s)|\pi_0(\cdot|s)) d_\rho^{\pi^*}(ds) \right.$$

$$\left. + d_1 \int_0^t e^{-\frac{\tau}{2}(t-r)} \frac{1}{\eta_r} dr \right).$$

### E.3 PROOF OF THEOREM 6.3

*Proof.* Following completely identically to the proof of Theorem 6.3, we have

$$\left| \frac{d\theta_{\pi_t}}{dt} \right| \leq \frac{\gamma}{\lambda_\beta(1-\gamma)} |A_\tau^{\pi_t}|_{B_b(S \times A)} \sup_{s \in S} |\partial_t \pi_t(\cdot|s)|_{\mathcal{M}(A)}$$

$$\leq \frac{4}{(1-\gamma)^2} \left( |c|_{B_b(S \times A)} + \mathrm{K}_t \right)^2 + 4\tau \left( C_1 + \frac{2}{\tau} \sup_{r \in [0,t]} |\theta_r| + \sup_{r \in [0,t]} \mathrm{K}_r \right)^2.$$

Then by Corollaries E.1 and E.2, there exists $b_2 > 0$ such that $\left| \frac{d\theta_{\pi_t}}{dt} \right|^2 \leq d_1$. Hence by Theorem 6.2 we have

$$\min_{r \in [0,t]} V_\tau^{\pi_r}(\rho) - V_\tau^{\pi^*}(\rho) \leq \frac{\tau}{2(1-\gamma)(1 - e^{-\frac{\tau}{2}})} \left( e^{-\frac{\tau}{2}t} \left( \int_S \mathrm{KL}(\pi^*(\cdot|s)|\pi_0(\cdot|s)) d_\rho^{\pi^*}(ds) \right. \right.$$

$$\left. \left. + d_1 \int_0^t e^{-\frac{\tau}{2}(t-r)} \frac{1}{\eta_r} dr \right).$$

$\square$

