# OpenReview forum: "Convergence of an actor-critic gradient flow for entropy regularised MDPs in general spaces"
_ICLR.cc/2026/Conference — ICLR 2026 Poster_

### Official Review · Reviewer_N1Ri · 2025-10-29

**Soundness:** 3
**Presentation:** 1
**Contribution:** 2
**Rating:** 4
**Confidence:** 3

**Summary:**

The paper studies infinite state-action space actor-critic with policy iteration and entropy regularization. It proves stability of the system and proves exponential convergence.

**Strengths:**

The analysis is done for continuous state-action space.

**Weaknesses:**

Presentation of results is bit cluttered, its little difficult to understand the contribution. See questions.

**Questions:**

Q1) Theorem 1.1 and Lemma 1.1 are insightful but I am not sure how novel it is. If it is established then I urge the authors to state it a proposition or property. And lemmas and theorems  be reserved for the results which are novel contribution of the paper.

Q2)  Policy Mirror Descent requires updating all states at each iteration, how can it be extended to online setting where we have access to only samples?

Q3) Corollary 5.1 (stabiliity): The constant $a_2$ >0, then how does the upper bound of $e^{a_2t}$ indicates stability, it is diverging very fast? Same for Corollary 5.2?

Q4) Corollary 5.3: At the limit $\tau \to\infty$, RHS converges to $a_1$. However, intuitively we expect that when the temperature becomes very high (so does the penality of deviating from the current policy) the policy should not change at all. In other words, at this limit $a_1$ should be zero. Is it the case?

Q5)  Authors mention that entropy regularization boosts exploration but policy iteration requires updating all states at each iteration. What benefits  entropy regularization provides to policy iteration?

Q6) In general, policy iteration conveniently convergence to the global optimal policy, and entropy regularization makes it even more stable ()

---

> ### Author Response · Authors · 2025-11-20
> **Response regarding weakness 1**
>
> We thank the reviewer for their valuable time and effort in providing detailed feedback on our work. We hope our response will fully address all of the reviewer's points.
>
> > "Presentation of results is bit cluttered, its little difficult to understand the contribution."
>
> We thank the reviewer for bringing this to our attention.
> We have restructured some of the paper by moving some known technical results to the appendix, such as the performance difference lemma and dynamic programming for entropy regularised MDPs.
>
> To address the main contribution of this paper, we refer the reviewer to our answer the Question 2 of Reviewer o1x9 where we explicitly state the main contirbutions of this work. Moreover, we have also added more detail in the introduction of the revised paper to communicate this point more clearly to the reader
>
> All revisions appear in the color cyan in the revised paper.

---

> > ### Author Response · Authors · 2025-11-20
> > **Response to questions 4.-6.**
> >
> > ## Question 4
> >
> > We thank the reviewer for this insightful remark and question.
> > The reviewer is correct, in the limit $\tau \to \infty$, the right-hand-side in the estimate in Corollary E.1 (used to be Corollary 5.1 pre-reivision) converges to $a_1$.
> > The reviewer is also correct that as $\tau \to \infty$ we would expect $\operatorname{KL}(\pi_t(\cdot|s)|\mu) \to 0$ and we have added a demonstration in Remark E.1 that our analysis can indeed achieve this.
> > However, Corollary E. 1 provides an estimate which, while not asymptotically tight, is the one that's sufficient for our analysis.
> >
> > ## Question 5
> > For the discussion regarding having to update all states at each iteration, we refer to our answer to Q2 about what would be done in practice and why we omit this.
> >
> > Let us recall the first term of Theorem 6.1, omitting constants:
> > $$
> > V^{\pi\_r}\_{\tau}(\rho) - V^{\pi^\*}\_{\tau}(\rho)  \leq e^{-\frac{\tau}{2} t} \int\_{S} \operatorname{KL}(\pi(\cdot|s)|\pi\_0(\cdot|s)) d_\rho^{\pi^\*}(ds) + \dots .
> > $$
> > Now suppose that we do not incorporate entropy regularisation in the value function (let $\tau \to 0$).
> > If the action space has finite cardinality, choosing the initial policy to be uniform, it directly holds that $\operatorname{KL}(\pi^\*(\cdot|s)|\pi_0(\cdot|s)) \leq \log |A|$ where $|A|$ represents the cardinality of the action space.
> >
> > In the continuous setting $\operatorname{KL}(\pi^\*(\cdot|s)|\pi_0(\cdot|s))$ is finite only if the density $\frac{d\pi^\*}{d\pi_0}$ exists.
> > However, by the dynamic programming principle, see e.g. [4] Theorem 4.2.3, (for $\tau = 0$) shows that the optimal policies will have support on a mixture of dirac distributions.
> > Therefore, $\operatorname{KL}(\pi^*(\cdot|s)|\pi_0(\cdot|s))$ will be finite only if $\pi_0$ is also a mixture of dirac distribution which contains the optimal policy.
> >
> > To paraphrase, in continuous action space you only get convergence of mirror descent if you set $\pi_0(\cdot|s)$ to have dirac mass at the same points as $\pi^\ast(\cdot|s)$ which is basically equivalent to knowing the optimal actions for each state.
> >
> > ## Question 6
> >
> > If the cardinality of the action space is finite then Fisher--Rao flow (mirror descent flow) with no entropy regularisation converge at rate of order $1/t$ while with entropy regularisation this is of order $e^{-\tau t}$.
> > Thus the entropy regularisation is seen to accelerate convergence.
> > For general action spaces one cannot omit the entropy regularisation as discussed above.
> > One could consider making $\tau$ depend on $t$ in a way that $\tau(t)\searrow 0$ as $\tau \to \infty$.
> > With additional regularity assumptions on the MDP one could show that such annealed flow leads to convergence to the value function of the unregularised problem.
> >
> > ## References
> > [1] Kerimkulov et al, A Fisher–Rao Gradient Flow for Entropy-Regularised Markov Decision Processes in Polish Spaces, Foundations of Computational Mathematics.
> >
> > [2] Haarnoja et al, Soft Actor-Critic: Off-Policy Maximum Entropy Deep Reinforcement Learning with a Stochastic Actor, International Conference on Machine Learning.
> >
> > [3] Kakade & Langford, Approximately Optimal Approximate Reinforcement Learning, International Conference on Machine Learning.
> >
> > [4] Howard, Dynamic Programming and Markov Processes, John Wiley.
> >
> > [5] Schulman et al, Trust Region Policy Optimization, International Conference on Machine Learning.
> >
> > [6] Schulman et al, Proximal Policy Optimization Algorithms, arXiv preprint arXiv:1707.06347.
> >
> > [7] Straughan, Explosive Instabilities in Mechanics, Springer.

---

> ### Author Response · Authors · 2025-11-20
> **Response to questions 1.-3.**
>
> ## Question 1
>
> We thank the reviewer for these remarks and we have made the relevant changes in the revised document.
>
> We agree with the reviewer that Theorem 1.1 is insightful.
> Indeed, it is the dynamic programming principle,  established in the 1950s by Bellman.
> The version we present in the paper holds for entropy regularised MDPs with general state and action spaces and the proof can be found in [1].
> As such it is not new.
> The purpose for presenting it was to see that the optimal policy has bounded log density and thus restricting to policies with bounded log densities is without loss of generality.
>
> Lemma 1.1 is the classical performance difference lemma going back to Howard [4], see also [3],  which was extended to the polish space + entropy regularised MDP setting in [1].
>
> For ease of reading and to make the presentation less cluttered, we have moved these results to the Appendix.
>
> ## Question 2
>
> Indeed in the current setting, we work under the theoretical conditions that the normalisation constant can be computed exactly and that we can update the policy for all states at each iteration. This is mainly since our focus is to establish the stability and convergence of the actor-critic flow in continuous action spaces and fill this gap in the literature.
>
> In practice, one can parametrize the policy and then use the right-hand-side of equation (3) (with the exact advantage replaced by the approximate one arising from the critic) to define a loss function which can be estimated from samples collected under the current policy.
> This loss is then be used with ADAM optimizer to update the actor weights.
> This is what is essentially what is done in TRPO [5] and PPO [6].
>
> Alternatively one can take equation (6), which is the exact mirror descent update, as a "target" and minimize a KL loss between this target and a policy arising from a parametrization of the actor.
> This is analogous to the approach taken by the SAC algorithm [2].
>
> To get better understand such algorithms in this setting, one should study discrete time stepping and sample complexity.
> This is an area of active ongoing research.
>
> ## Question 3
> We thank the reviewer for this important question. To clarify, we mean stability in the sense of no finite time blow up, that is there does not exist $T_{\max} < \infty$ such that $\lim_{t \to T_{\text{max}}}\operatorname{KL}(\pi_t(\cdot|s) | \mu) = \infty$.
>
>
> Existence of such a time $T_{\text{max}}$ would result in a singularity in the actor critic dynamics. The stability results in Corollary 5.1 and 5.2 therefore show that under the assumptions and setting of this work, the entropy term does not suffer from a finite-time blow-up.
> See e.g [7] for a comprehensive treatment and examples of dynamics with finite time blow-up.
> Moreover, by following the proof of Theorem 6.3, we show that the exponential bounds arising from Corollaries 5.1 and 5.2 can be controlled through $\eta_t$, the critic speed.

---

> ### Author Response · Authors · 2025-11-28
>
> Dear Reviewer N1Ri,
>
> Thank you once again for your valuable review, it has undoubtedly helped us improve the manuscript  and we have uploaded the revised version.
>
> As the author-reviewer discussion phase is coming to an end, we hope that you can find the time to review our responses to your questions and concerns.
>
> Best wishes,\
> Authors.

---

### Official Review · Reviewer_354V · 2025-10-30

**Soundness:** 3
**Presentation:** 3
**Contribution:** 2
**Rating:** 6
**Confidence:** 3

**Summary:**

The paper analyzes an actor-critic method in continuous time with (i) TD learning as the critic, and (ii) policy mirror descent for policy optimization to solve entropy-regularized MDPs with general (Polish) state-action spaces. Under a realizability assumption on Q-functions throughout the trajectory, and for sufficiently small $\gamma$, an exponential convergence rate up to a critic error was established.

**Strengths:**

- The paper is very well-written and mathematically rigorous. The technical contribution is solid.
- Convergence of an actor-critic method based on TD-critic and PMD-actor was analyzed for MDPs with general (Polish) state-action spaces. The existing works analyze entropy-regularized actor-critic methods for finite action spaces, where the optimality gap has a term $\sqrt{\frac{\log |A|}{t}}$. Since this bound becomes vacuous for continuous action spaces, the contribution of this paper is significant.
- The paper establishes a stability analysis that explicitly captures the interplay between entropy regularization and timescales.

**Weaknesses:**

- The paper makes a strong assumption that $\gamma$ should be sufficiently small, given that the practical $\gamma$ values are typically larger than 0.9. Indeed, the range of $\gamma$ shrinks as $\tau$ increases (e.g., in Corollary 6.1).
- Equation (40) indicates that the decay rate of the optimality gap is $O(t^{-1/2})$. This makes the exponential convergence argument in the abstract a little confusing. I would recommend revising that statement in the abstract to indicate that it is exponential convergence up to an additional error term that stems from the critic.
- The impact of stochasticity is not characterized as the analysis is fully deterministic. For TD learning, this impact can be particularly significant, and may have significant implications for policy optimization.
- The realizability assumption is strong, but it can be relaxed by including an additional projection error term.
- A continuous action space $A$ would make the computation of an exact softmax policy intractable due to the integration in Line 188-189. As such, in order to sample from this policy, one may either compute an inexact policy (via an approximate integral), or use Langevin dynamics to sample from $\pi$ without explicitly computing it. In any case, this brings an additional error term. For this reason, in practice, this extension of finite-action softmax can be intractable and variants such as deterministic policy gradient (Silver et al., 2014) are used.
- (Minor) There is a continual switching between "regularisation" and "regularization". I recommend making it consistent.

**References**

Silver, D., Lever, G., Heess, N., Degris, T., Wierstra, D., & Riedmiller, M. "Deterministic Policy Gradient Algorithms." In International Conference on Machine Learning (pp. 387-395), PMLR, 2014.

**Questions:**

- An interesting connection between critic error and the policy optimization was established in (Kakade, 2001), which is known as the compatible function approximation. One can formulate the NPG updates as the minimizer of $L_\theta(w) = E_{s,a}[|\nabla \log \pi_\theta(a|s) w - A^{\pi_\theta}(s,a)|^2]$. For linear function approximation, this may roughly be thought as minimizing $E|\phi({s,a})\cdot w - Q^{\pi_\theta}(s,a)|^2$. Since one also uses TD learning with linear function approximation using the same feature map $\phi({s,a})$ to approximate $Q^{\pi_\theta}$, which gives us an exact solution under the realizability assumption, the insight from (Kakade, 2001) also applies here nicely. Is this idea used in the paper in an related/alternative form? The joint Lyapunov analysis that connects the actor and critic may possibly have a connection with this.
- In conjunction with the above point, the realizability seems to be a significant contributor to the exponential convergence rate. What would happen without this realizability assumption? More clearly, how would the projection error propagate?
- Is the small-$\gamma$ assumption inevitable? Additional discussion on its cause would be useful.

**References**

Kakade, S. M. "A Natural Policy Gradient." Advances in Neural Information Processing Systems 14, 2001.

---

> ### Author Response · Authors · 2025-11-20
> **Response to first batch of questions and weaknesses**
>
> We thank the reviewer for taking the time to carefully read our submission. We have uploaded a revision of the manuscript where the changes made are highlighted in cyan.
>
> ## Responses
>
> >The paper makes a strong assumption that $\gamma$ should be sufficiently small, given that the practical  $\gamma$ values are typically larger than 0.9.
>
> We would like to emphasise that the small $\gamma$ condition is merely a special case and that the main stability and convergence results of this paper: Theorem 5.1, Corollaries 5.1, 5.2 and Theorems 6.1, 6.2 and 6.3 hold for all $\gamma \in (0,1)$.
>
> To ease readability we moved the small $\gamma$ special case results to the appendix in the revised version of the manuscript.
>
>
> > Equation (40) indicates that the decay rate of the optimality gap is
> $\mathcal{O}\left(t^{-{\frac{1}{2}}}\right)$. This makes the exponential convergence argument in the abstract a little confusing. I would recommend revising that statement in the abstract to indicate that it is exponential convergence up to an additional error term that stems from the critic.
>
> We thank the reviewer for this important remark. Firstly, Theorem 6.3 shows that for all $\gamma \in (0,1)$, if the critic speed is fast enough we can still maintain the exponential convergence of the actor dynamics without solving the critic to full accuracy at each iteration.
>
> Corollary E.3 is then a special case of this result which shows that if the effective horizon of the MDP is short enough due to sufficiently small discount factor, then we have more control on the convergence rate by running the critic flow slower. The $\mathcal{O}\left(t^{-{\frac{1}{2}}}\right)$ is then an example of this special case.
>
> We have moved these special case for small $\gamma$ to the Appendix.
> We also made the reviewer's recommanded changes in the abstract and introduction.
>
> >The impact of stochasticity is not characterized as the analysis is fully deterministic. For TD learning, this impact can be particularly significant, and may have significant implications for policy optimization.
>
> We agree with the reviewer and in fact we've pointed this out as a shortcoming in the Limitations section.
> Since we are focusing on rigorously treating how the entropy impacts stability and convergence in the continuous action space setting under minimal assumptions, we assume that all integrals can be calculated exactly and thus the stochasticity from TD learning is not addressed and is the result of ongoing research.
>
> >The realisability assumption is strong, but it can be relaxed by including an additional projection error term.
>
> We will comment on this along with how the error may propagate along the flow.
> We also refer the reviewer to our respond for question 2 of reviewer o1x9.
>
> >A continuous action space $A$
> would make the computation of an exact softmax policy intractable due to the integration in Line 188-189. As such, in order to sample from this policy, one may either compute an inexact policy (via an approximate integral), or use Langevin dynamics to sample from
> without explicitly computing it. In any case, this brings an additional error term. For this reason, in practice, this extension of finite-action softmax can be intractable and variants such as deterministic policy gradient (Silver et al., 2014) are used.
>
> We agree with the reviewer that this is a major challenge in practice.
> A common solution is to only parametrize mean and covariance of normal distributions whereby sampling becomes computationally trivial.
> Let's say these parameters are $\psi \in \mathbb R^{N'}$ leading to parametrized policy $\pi_{\psi}$.
> One the updates these parameters by treating the the mirror descent update (3) as a target and one minimizes $\operatorname{KL}(\pi_\psi| \pi^{n+1})$ with a gradient algorithm, similarly to what is done in SAC [2].

---

> ### Author Response · Authors · 2025-11-20
> **Response to second batch of questions and weaknesses**
>
> > An interesting connection between critic error and the policy optimization was established in (Kakade, 2001), which is known as the compatible function approximation. One can formulate the NPG updates as the minimizer of $L\_{\theta}(w) = \mathbb{E}\_{s,a} \left( \left| \nabla \log \pi_{\theta}(a|s) w - A^{\pi_\theta}(s,a)\right|^2 \right)$. For linear function approximation, this may roughly be thought as minimizing $\mathbb{E} |\phi(s,a) \cdot w - Q^{\pi_{\theta}}(s,a)|^2.$. Since one also uses TD learning with linear function approximation using the same feature map $\phi(s,a)$ to approximate $Q^{\pi_{\theta}}$, which gives us an exact solution under the realisability assumption, the insight from (Kakade, 2001) also applies here nicely. Is this idea used in the paper in a related/alternative form? The joint Lyapunov analysis that connects the actor and critic may possibly have a connection with this.
>
> This is a good observation.
> In NPG one would take log-linear policies and consider the loss
> $$
> L^{\pi\_\theta}(w):=\frac12 \int\_S\int\_A |A^{\pi\_\theta}\_\tau(s,a) - \langle w, \phi\_{\pi\_\theta}(s,a) \rangle|^2\pi\_\theta(da|s)d\_\rho^{\pi\_\theta}(ds)\,,
> $$
> which is exactly what the loss above with the expectation being made more explicit and where $\phi_{\pi_\theta}$ are the centered features (their mean over $A$ under current policy $\pi_\theta$ subtracted).
> Let $\hat w(\theta)$ be the unique minimizer of the above loss.
> Under the compatible function approximation (essentially $Q^\pi_\tau$ realisability) the NPG updates
> $$
> \theta_{n+1} = \theta_n - \tfrac{\eta}{1-\gamma} \big(	\hat w(\theta_n)+\tau \theta_n\big)\,,\,\,\,n=0,1,\ldots\,,\,\,\, \theta_0 \in \mathbb R^N \,\,\,\text{given}
> $$
> leading to policies with log-densities $\ln \tfrac{d\pi^n}{d\mu} = \langle\theta^n, \phi\rangle$
> is equivalent to the mirror descent updates (3) where the exact advantage function is replaced by its least-squares approximation given by $\langle \hat w(\omega), \phi \rangle$.
>
> The connection to the actor critic flow is that the $\hat w(\omega)$ would be the limit of the critic flow if one kept the policy $\pi$ fixed.
> However, the analysis of the NPG is much simpler because, under $Q^\pi_\tau$ realisability the NPG improves the value function and thus one can show that the $\operatorname{KL}$ remains bounded along the NPG updates.
> Even without this assumption the convergence analysis (assuming no sampling errors i.e. exact integral evaluations) is relatively straightforward and can be found e.g. in [1].
>
> >Is the small-$\gamma$ assumption inevitable? Additional discussion on its cause would be useful.
>
> The short answer is *no*.
> The paper presents results for all $\gamma \in (0,1)$ but at the cost of needing a critic timescale $\eta_t \sim e^t$.
>
> In the appendix we present the special case where $\eta_t$ can be e.g. of order $t^{1/2}$ but then one needs sufficiently small $\gamma$.
> We agree with the reviewer that in practice $\gamma$ close to $1$ is more relevant which is why we removed the small $\gamma$ case from the main text.
>
> The small $\gamma$ condition arises due to the coupled analysis, entropy regularisation and the continuous action space setting. It acts as a way of enforcing extra regularity on the MDP to achieve more regularity of the policies and critic parameters.
>
> Interestingly, similar phenomenon for stability has also been observed in [3,4].
>
> >In conjunction with the above point, the realisability seems to be a significant contributor to the exponential convergence rate. What would happen without this realisability assumption? More clearly, how would the projection error propagate?
>
> Let us denote best approximation error as $\delta\_\pi := |Q^\pi\_\tau - \langle \theta\_\pi, \phi \rangle|\_{B\_b(S\times A)}$, where $\theta\_\pi = \text{argmin}\_\theta  |Q^\pi\_\tau - \langle \theta, \phi \rangle|\_{B\_b(S\times A)}$.
> This will give rise to an additional error term $\frac{1}{\tau}\int_0^t e^{-\tau(t-r)} \delta_r dr$ on the right hand side of the main convergence results which is displayed in Theorem 6.3.
> However, all the steps in the proofs will need to be checked carefully.
> Together with dealing with inexact integral evaluations (sampling errors), we leave this analysis for future research.
>
> [1] Kerimkulov et al, A Fisher–Rao Gradient Flow for Entropy-Regularised Markov Decision Processes in Polish Spaces, Foundations of Computational Mathematics.
>
> [2] Haarnoja et al, Soft Actor-Critic: Off-Policy Maximum Entropy Deep Reinforcement Learning with a Stochastic Actor, International Conference on Machine Learning.
>
> [3] Devraj & Meyn, Zap Q-Learning, Advances in Neural Information Processing Systems.
>
> [4] Meyn, The Projected Bellman Equation in Reinforcement Learning, IEEE Transactions on Automatic Control.

---

> ### Author Response · Authors · 2025-11-28
>
> Dear Reviewer 354V,
>
> Thank you once again for your valuable review, it has undoubtedly helped us improve the manuscript  and we have uploaded the revised version.
>
> As the author-reviewer discussion phase is coming to an end, we hope that you can find the time to review our responses to your questions and concerns.
>
> Best wishes,\
> Authors.

---

### Official Review · Reviewer_Va3v · 2025-10-30

**Soundness:** 3
**Presentation:** 3
**Contribution:** 3
**Rating:** 8
**Confidence:** 3

**Summary:**

The paper proves the convergence of an actor-critic algorithm for entropy-regularized reinforcement learning (RL) in the setting with continuous states, actions and time dynamics. Concretely, the paper shows that the algorithm is stable and converges at an exponential rate.

**Strengths:**

If the proofs are correct and novel, I believe that the paper makes an important contribution.

**Weaknesses:**

I am well familiar with the theory of entropy-regularized RL, but I am not an expert on continuous-time dynamics, which makes some of the proofs difficult to follow.

**Questions:**

The authors choose to regularize on a fixed action distribution \mu rather than on a full stochastic policy that selects a different action distribution in each state. Several existing algorithms such as TRPO regularize on full policies, so this seems like a limiting choice. Why do you not regularize on a full policy, and would it be difficult to extend the result to such policies?

What do you mean by a policy being "equivalent" to the reference action distribution \mu? My interpretation would be that the policy selects the same action distribution \mu in each state, but I believe that this is not what you mean.

Why did you choose to minimize the MSBE rather than some other objective? There are other objectives that seem to have better properties for entropy-regularized RL, e.g. Logistic Q-learning.

Equation (23) is a non-regularized objective, so Lemma 4.1 seems to imply that we can upper bound the regularized gradient by a non-regularized gradient. This is curious to me since I thought that bounding the regularized gradient directly would be easier. Why did you choose this particular proof strategy?

Lemma 5.1 looks familiar to me as a flow constraint, but I have a hard time interpreting Lemma 5.2, which looks like a differential equation. What is the interpretation of this lemma?

---

> ### Author Response · Authors · 2025-11-20
> **Response to questions and weaknesses**
>
> We thank the reviewer for taking the time to carefully read our submission.
> We have uploaded a revision of the manuscript where the changes made are highlighted in cyan.
>
> ### Question 1
>
> > The authors choose to regularize on a fixed action distribution $\mu$ rather than on a full stochastic policy that selects a different action distribution in each state. Several existing algorithms such as TRPO regularize on full policies, so this seems like a limiting choice. Why do you not regularize on a full policy, and would it be difficult to extend the result to such policies?
>
> *Response:*
> We thank the reviewer for this question and for the opportunity to clarify the role of regularisation in our work.
>
> First, in the entropy-regularised MDP we define, for each stochastic policy $\pi \in \mathcal{P}(A|S)$ we have
> $$
> V^{\pi}\_{\tau}(s) = \mathbb{E}\_{s}^{\pi} \Bigg[\sum_{n=0}^\infty\gamma^n \Big(c(s\_n,a\_n) + \tau \operatorname{KL}(\pi(\cdot|s\_n)|\mu)\Big)\Bigg].
> $$
> The presence of the entropy term in the value function penalises the policies $\pi$ that deviate from some fixed reference measure $\mu$.
> This used to encourage random actions and thus exploration by the agent.
> In the current continuous action sapce setting it is usually taken to be of the form $\mu(da) \propto e^{-U(a)} da$ such that we have full support on the action space $A$.
>
> Second, if one is doing discrete stepping then in the mirror descent step
> $$
> \pi^{n+1}(\cdot|s)
> = \text{argmin}\_{m \in \mathcal{P}(A)}\left[ \int_{A} A(s,a,\theta^{n+1})\big(m(da) - \pi^{n}(da|s)\big) + \frac{1}{\lambda}\operatorname{KL}(m|\pi^{n}(\cdot|s))\right],
> $$
> the KL term $\operatorname{KL}(m|\pi^n(\cdot|s))$ is a proximal regularizer on the update of the policy, penalising deviations of $\pi^{n+1}(\cdot|s)$ from the previous $\pi^n(\cdot|s)$. This plays the same conceptual role as the trust-region regularizer in TRPO.
>
> ### Question 2
>
> > What do you mean by a policy being "equivalent" to the reference action distribution $\mu$? My interpretation would be that the policy selects the same action distribution $\mu$ in each state, but I believe that this is not what you mean.
>
> *Response:*
> Apologies, we should have defined the term. To be clear, we say that two measures $\mu, \nu \in \mathcal{P}(A)$ are equivalent if they are mutually absolutely continuous, i.e. $\mu \ll \nu$ and $\nu \ll \mu$.
> To avoid confusing other readers we've removed this term from the main text.
>
> ### Question 3
> > Why did you choose to minimize the MSBE rather than some other objective? There are other objectives that seem to have better properties for entropy-regularised RL, e.g. Logistic Q-learning.
>
> *Response:*
> We thank the reviewer for this interesting question and insight.
> Firstly, the MSBE is a classical objective in both in the unregularised and entropy-regularised settings, making our analysis directly comparable to the existing literature.
>
> Moreover, to control the critic dynamics our analysis relies on the fact that the descent direction of the semi gradient of the mean-squared Bellman error aligns with the true gradient of the squared loss.
> This is in turn a consequence of the flow constraint in Lemma A.1 in the revised manuscript and is characterized by Lemma 4.1.
> We refer the reviewer to the answer of the next question for a more detailed discussion on how this is used in the proof.
>
> Nevertheless, it would be interesting to analyze similar coupled flows with different objective functions for the critic and we leave this for future work.
>
> ### Question 4
>
> > Equation (23) is a non-regularised objective, so Lemma 4.1 seems to imply that we can upper bound the regularised gradient by a non-regularised gradient. This is curious to me since I thought that bounding the regularised gradient directly would be easier. Why did you choose this particular proof strategy?
>
> *Response:*
> We apologies but it seems that in Definition 4.1  we missed the subscript $\tau$ leading to the impression that this is the unregularised $Q$ function.
> This has been fixed in the revised version.
>
> ### Question 5
>
> > Lemma 5.1 looks familiar to me as a flow constraint, but I have a hard time interpreting Lemma 5.2, which looks like a differential equation. What is the interpretation of this lemma?
>
> *Response:*
> Lemma 5.1 (in the original submission, moved to Lemma A.1 in the revision) is indeed a flow constraint for the state-action occupation measure.
> Lemma 5.2 (in the original submission, now Lemma 5.1) illustrates that the coupled actor-critic flow is a forcing-damping system and is the first step towards showing that the coupled actor-critic system does not suffer from a finite time blow up.
> We have added comments to that end in the main text, see top of Section 5 and also directly after Lemma 5.1 (in the revised version).

---

> ### Author Response · Authors · 2025-11-28
>
> Dear Reviewer Va3v,
>
> Thank you once again for your valuable review, it has undoubtedly helped us improve the manuscript  and we have uploaded the revised version.
>
> As the author-reviewer discussion phase is coming to an end, we hope that you can find the time to review our responses to your questions and concerns.
>
> Best wishes,\
> Authors.

---

### Official Review · Reviewer_o1x9 · 2025-10-31

**Soundness:** 3
**Presentation:** 3
**Contribution:** 2
**Rating:** 4
**Confidence:** 3

**Summary:**

This paper presents theoretical analyses of the Actor-Critic algorithm for continuous action spaces in entropy-regularized Markov Decision Processes (MDPs). In addition, it provides explicit gradient expressions for both the critic and actor updates.

**Strengths:**

1. The theoretical results presented in this work appear to be sound and logically consistent with the proposed framework. The derivations are coherent, and the assumptions, although somewhat idealized, support the validity of the main conclusions. Overall, the theoretical analysis contributes to a solid understanding of the algorithm’s behavior and convergence properties.

2. In addition, the definitions and problem descriptions are generally clear and well-structured. The authors provide sufficient background and notation, making the paper accessible to readers familiar with reinforcement learning and KL-regularized optimization. The clarity of exposition facilitates understanding of both the motivation and the technical development of the method.

**Weaknesses:**

1. Some assumptions, such as Assumption 4.1, seem overly strong and may not hold in practical continuous or high-dimensional settings. The convergence results appear to rely heavily on these assumptions.


2. The technical novelty and improvements over prior work are unclear, and the contribution would benefit from a clearer explanation of how it advances existing methods.

**Questions:**

1. Assumption 4.1 appears to be quite strong and may not hold in practical scenarios, particularly in environments with continuous state and action spaces (potentially high-dimensional). Does the convergence guarantee in this work critically rely on this assumption?

2. Could the authors elaborate on the technical challenges encountered when extending the analysis or algorithm to continuous state and action spaces? How were these challenges addressed in the proposed framework?

---

> ### Author Response · Authors · 2025-11-20
> **Response to question and weakness 1**
>
> We thank the reviewer for taking the time to carefully read our submission. We have uploaded a revision of the manuscript where the changes made are highlighted in cyan.
>
> ## Question 1
>
> We will address the use of Assumption 4.1 ($Q^{\pi}_{\tau}$ realisability) in two different manners: i) its benefits in analysis and how it can be relaxed and ii) the practicality of such an assumption in continuous settings.
>
> i) Regarding the first point, the purpose of $Q^{\pi}_{\tau}$ realisability is to enable us to perform analysis which would be intractable under richer, non-linear function approximations. For this reason, realisability has been used in several seminal and recent works [4,5,6,7].
>
> The realisability assumption is in-fact not used in the stability analysis as the temporal difference (TD) semi-gradient flow (8) does not require knowledge of the true $Q^{\pi}_{\tau}$.
>
> As the reviewer observed, realisability is required in order to achieve a full convergence to the optimal policy.
> In principle, the $Q^\pi_\tau$ realisability assumption can be relaxed by including an additional error term in the analysis.
> However, in practice, one cannot easily quantify the error term and thus we feel this would be of limited utility and would come at expense of readability of the overall argument.
>
>
>
> ii) To address the practicability of $Q^{\pi}\_{\tau}$ realisability, consider the following example:
> Suppose that $S = \mathbb{R}^d$ and consider the following state dynamics $s_{t+1} = f(s_t,a_t) + \epsilon_t$ for some continuous function $f : S \times A \to S$ and $\epsilon_t \sim N(0,\sigma^2 I_{d})$.
> Clearly the transition kernel satisfies
> $$
> P(s'|s,a) \propto \exp\left(- \frac{|f(s,a) - s'|\_2^2}{2\sigma^2} \right).
> $$
> Then by Bochner's Theorem there is a basis $(\phi\_n)\_{n\in \mathbb N}$ and signed measures on $S$ denoted $(\mu\_n)\_{n \in \mathbb N}$ such that $P(ds'|s,a) = \sum_{n\in N}\phi(s,a) \mu(ds')$.
> This however needs infinitely many basis functions, therefore careful truncation of the basis functions is required for practical purposes.
> This approach has had empirical success for Quadrotor control [2] and Pendulum SwingUp [3] tasks.
>
>
> We have included some extra discussions about this assumption in the revised document to help the reader understand the purpose of this assumption.

---

> ### Author Response · Authors · 2025-11-20
> **Response to question and weakness 2**
>
> ## Question 2
>
> We thank the reviewer for this important question.
> Thus far, the convergence and stability properties of TD-learning + Policy Mirror Descent actor critic for entropy regularised MDPs in continuous action spaces was an open question.
> Existing works rely heavily on the finite cardinality of the action space: in such a setting the entropy term is upper bounded by a constant depending on the cardinality of the action space independent of the policies (see [1]) for any $\mu$ such that $\mu(a_i)>0$, $i=1,2,\ldots,|A|$.
> In this setting all the relevant objects ($Q^\pi_\tau$ etc.) are bounded uniformly by a constant depending on the cost, discounting and cardinality of $A$.
>
> For general action spaces, this no longer holds and thus new analytical techniques are required and developed in this paper.
>
> Moreover, even in the finite cardinality of the action space setting, existing results often employ clipping of the critic parameters to ensure stability.
> Additionally, other than $Q^{\pi}_{\tau}$ realisability, this work operates under minimal assumptions.
> Unlike prior results, we do not require concentrability coefficients (which may not be finite in practice) nor other regularity assumptions on the policies [1,8].
>
> Furthermore, our analysis is carried out for general Polish state and action spaces. To the best of our knowledge, this is among the most general settings in which actor-critic methods have been studied so far. This level of generality is particularly useful for partially observed control problems: a POMDP with a Polish hidden state space can be reformulated as a fully observed "belief MDP", whose state space is the space $\mathcal{P}(S)$ of probability measures on the hidden state space endowed with the weak topology. The space $\mathcal{P}(S)$ is itself Polish, so our assumptions cover such infinite-dimensional belief-state models, in addition to the usual finite- and continuous-state MDPs. See [9] for a detailed discussion and review.
>
>
> Our work addresses this gap in the literature which we believe is important for the communities understanding on the foundations of reinforcement learning and assisting in algorithm design.
>
>
>
>
>
>
>
>
> ## References
>
> [1] Cayci et al, Convergence of Entropy-Regularized Natural Policy Gradient with Linear Function Approximation, SIAM Journal on Optimization. \
> [2] Ma et al, Skill Transfer and Discovery for Sim-to-Real Learning: A Representation-Based Viewpoint, International Conference on Intelligent Robots and Systems. \
> [3]Ren at al, Stochastic Nonlinear Control via Finite-dimensional Spectral Dynamic Embedding, Proceedings of the 2023 62nd IEEE Conference on Decision and Control.\
> [4] Bochner, Harmonic Analysis and the Theory of Probability, University of California Press.\
> [5] Bhandari et al, A Finite Time Analysis of Temporal Difference Learning with Linear Function Approximation, Operations Research.\
> [6] Tkachuk et al, Trajectory Data Suffices for Statistically Efficient Learning in Offline RL with Linear $q_{\pi}$-Realizability and Concentrability, Advances in Neural Information Processing Systems.\
> [7] Meyn, The Projected Bellman Equation in Reinforcement Learning, IEEE Transactions on Automatic Control. \
> [8] Hong et al, A Two-Timescale Stochastic Algorithm Framework for Bilevel Optimization: Complexity Analysis and Application to Actor-Critic, SIAM Journal on Optimization.
> [9] Kara et al, Q-Learning for MDPs with General Spaces: Convergence and Near Optimality via Quantization under Weak Continuity, Journal of Machine Learning Research.

---

> ### Author Response · Authors · 2025-11-28
>
> Dear Reviewer o1x9,
>
> Thank you once again for your valuable review, it has undoubtedly helped us improve the manuscript  and we have uploaded the revised version.
>
> As the author-reviewer discussion phase is coming to an end, we hope that you can find the time to review our responses to your questions and concerns.
>
> Best wishes,\
> Authors.

---

### Meta-Review · Area_Chair_DYWp · 2026-01-07

**Summary:**

While reviewers generally agreed on the technical soundness and ambition of the paper, several common concerns were raised across the reviews:

1. Strength and Practicality of Assumptions:
   Multiple reviewers noted that the analysis relies on strong assumptions, most notably Q-function realizability and, in some regimes, restrictive conditions on the discount factor or critic timescale. Reviewers questioned how realistic these assumptions are in high-dimensional continuous state–action settings and to what extent the convergence guarantees depend critically on them. Relatedly, some reviewers expressed concern that practical implementations would introduce additional approximation or sampling errors not captured in the analysis.

2. Practical Relevance and Implementability:
   Reviewers highlighted that the results are derived in a fully deterministic, continuous-time setting with exact integral evaluations. In particular, concerns were raised about the tractability of computing or sampling from entropy-regularized policies in continuous action spaces, and about the absence of stochastic or sample-based analysis. While these limitations were acknowledged as outside the scope of the paper, they affected some reviewers’ assessment of practical impact.

3. Clarity of Novelty Relative to Prior Work:
   A few reviewers found it initially unclear how the main results advance beyond existing convergence analyses for entropy-regularized actor–critic methods, especially in relation to finite-action or natural policy gradient settings. Some early sections and results were perceived as standard or classical, and reviewers suggested that the distinction between known background material and novel contributions be made more explicit.

4. Presentation and Accessibility:
   Several reviewers commented that the paper is technically dense and, at times, difficult to follow, particularly for readers less familiar with continuous-time gradient flows. There were concerns about cluttered presentation, overloaded notation, and insufficient high-level intuition in parts of the analysis, which made it harder to appreciate the core contributions.

5. Interpretation of Stability and Convergence Results:
   Questions were raised regarding the interpretation of stability bounds, especially where upper bounds grow exponentially in time, and how these results should be understood in relation to intuitive notions of stability. Additionally, some reviewers sought clearer explanations of the interaction between entropy regularization, timescale separation, and the resulting convergence rates.

Overall, the reviewers viewed these concerns as affecting clarity, scope, and practical relevance rather than indicating fundamental flaws in correctness, but they contributed to variability in enthusiasm and confidence across the reviews.

**Reviewer Concerns:**

The authors provided detailed and thoughtful rebuttals and revised the manuscript accordingly. Several key concerns raised by the reviewers were satisfactorily addressed, while a smaller number remain inherent limitations of the current work.

**Concerns Satisfactorily Addressed**

1. Role and Necessity of the Realizability Assumption:
   The rebuttal clearly clarified where realizability is essential and where it is not. In particular, the authors explained that:

   * Realizability is not required for the stability (non–finite-time blow-up) results.
   * It is only needed to obtain full convergence to the optimal regularized policy.
   * Without realizability, an additional projection error term would appear and propagate through the analysis.
     This clarification, together with added discussion in the revised manuscript, satisfactorily addresses reviewers’ questions regarding the logical role of this assumption, even if it remains strong.

2. Discount Factor and “Small-$\gamma$” Regime:
   Concerns regarding restrictive assumptions on the discount factor were addressed by restructuring the paper. The authors clarified that:

   * The main stability and convergence results hold for all discount factors, provided sufficient critic timescale separation.
   * Results requiring small discount factors are special cases and have been moved to the appendix.
     This substantially reduces confusion about the scope of the main theorems and resolves a major source of reviewer concern.

3. Clarity on Stability and Finite-Time Blow-Up:
   Reviewers questioned how exponentially growing bounds could be reconciled with stability. The rebuttal convincingly clarified that:

   * Stability is defined in the sense of absence of finite-time blow-up.
   * The exponential bounds serve as sufficient conditions for global existence of the flow and can be controlled via critic speed.
     Additional explanations added to the revised manuscript improve the interpretability of these results.

4. Novelty Relative to Prior Work:
   The authors more clearly articulated how their results go beyond existing work limited to finite action spaces, bounded entropy terms, or stronger assumptions such as concentrability. The rebuttal emphasizes that:

   * Unbounded entropy in continuous action spaces introduces genuinely new analytical challenges.
   * The stability analysis itself is novel in this setting.
     This clarification addresses concerns about overlap with prior literature and makes the contribution more explicit.

5. Terminology, Technical Errors, and Presentation Issues:
   Several minor but potentially confusing issues—such as unclear terminology (e.g., “equivalent measures”), notation errors, and mixing of classical and novel results—were corrected. Moving standard results to the appendix and adding signposting improved overall readability.

**Concerns Still Outstanding**

1. Practical Implementability and Sampling Issues:
   While the authors acknowledged the difficulty of computing or sampling from entropy-regularized policies in continuous action spaces and discussed common approximations (e.g., parametric policies, Langevin dynamics), these issues remain unresolved within the theory. The analysis still assumes exact integral evaluations and does not quantify errors arising from approximate sampling or policy parameterization.

2. Absence of Stochastic or Discrete-Time Analysis:
   The work remains entirely deterministic and formulated in continuous time. The impact of stochastic TD updates, finite samples, and discrete-time implementations—central to practical actor–critic algorithms—is explicitly left for future work. Although this limitation is clearly acknowledged, it remains a gap relative to applied relevance.

3. Strength of Realizability for Practical Settings:
   Although its role is now well explained, realizability itself remains a strong assumption, especially in high-dimensional or nonlinear function approximation regimes. The rebuttal clarifies how errors would enter the analysis but does not provide guarantees under relaxed conditions.

4. Accessibility for a Broader Audience:
   Despite improvements, the paper remains technically dense and challenging for non-specialists, particularly those unfamiliar with gradient flows or measure-theoretic RL. This is not a correctness issue but may limit accessibility for a broader ICLR audience.

**Overall Assessment**

The rebuttal satisfactorily addressed the core technical and conceptual concerns raised by reviewers, particularly regarding assumptions, scope, and novelty. The remaining issues are largely acknowledged limitations rather than unresolved flaws, and primarily concern practical applicability and scope rather than theoretical soundness.

**Reviewer Scores:**

Reviewer o1x9 => **From 4 → 5 or 6**

Primary concerns:

* Strength and practicality of Assumption 4.1 (realisability)
* Unclear technical novelty over prior work
* Limited intuition for continuous spaces

Rebuttal impact:
The rebuttal directly addressed all of this reviewer’s major concerns:

* Clearly separated stability results from convergence results, showing realizability is not needed for stability.
* Explained how realizability could be relaxed at the cost of an additional error term.
* Explicitly articulated novelty relative to finite-action-space analyses and bounded-entropy settings.
* Added clarifying discussion in the manuscript.

The reviewer already indicated they “would not mind if the paper is accepted,” suggesting openness.

---

Reviewer Va3v => **Remains at 8**

Primary concerns:

* Conceptual questions about regularization choice
* Clarifications of terminology
* Interpretation of certain lemmas

Rebuttal impact:
All questions were answered cleanly and constructively:

* Regularization choice was clarified conceptually and technically.
* Terminology issues were acknowledged and fixed.
* Proof intuition and interpretation were expanded in the revised text.

This reviewer already viewed the paper positively and framed concerns as clarification-oriented rather than critical.

---

Reviewer 354V => **From 6 → 6 or 7**

Primary concerns:

* Small-(\gamma) assumption and confusion around exponential convergence
* Practical intractability of continuous-action softmax
* Deterministic analysis ignoring stochastic TD errors
* Strength of realizability assumption

Rebuttal impact:
The rebuttal made meaningful improvements:

* Clarified that small-(\gamma) is a special case and moved it to the appendix.
* Revised the abstract to more accurately reflect convergence “up to critic error.”
* Acknowledged stochasticity and sampling issues as limitations.
* Provided a detailed discussion connecting realizability to compatible function approximation and NPG.

While practical concerns remain, the reviewer’s main confusion about scope and claims was resolved.


---

Reviewer N1Ri => **From 4 → 5**

Primary concerns:

* Cluttered presentation and unclear contribution
* Questions about novelty of early results
* Confusion around stability bounds and interpretation
* Conceptual questions about entropy regularization and policy iteration

Rebuttal impact:
The authors:

* Moved classical results to the appendix.
* Explicitly identified which results are standard vs. novel.
* Clarified the definition of stability as absence of finite-time blow-up.
* Addressed conceptual misunderstandings regarding entropy regularization limits.

However, this reviewer appeared less comfortable with the theoretical framing overall, and some concerns stem from expectations about online or sample-based algorithms that are outside the paper’s scope.

---

### Decision · Program_Chairs · 2026-01-26

Accept (Poster)